



# Vertical cloud structure of warm conveyor belts – a comparison and evaluation of ERA5 reanalyses, CloudSat and CALIPSO data

Hanin Binder[1], Maxi Boettcher[1], Hanna Joos[1], Michael Sprenger[1], and Heini Wernli[1]

[1]Institute for Atmospheric and Climate Science, ETH Zurich, 8092 Zurich, Switzerland

**Correspondence:** Hanin Binder (hanin.binder@env.ethz.ch)

**Abstract.** Warm conveyor belts (WCBs) are important cyclone-related airstreams that are responsible for most of the cloud and precipitation formation in the extratropics. They can also substantially influence the dynamics of the cyclone and the upper-level flow. So far, most knowledge about WCBs is based on model data from analyses, reanalyses and forecast data, with only a few observational studies available. The aim of this work is to gain a detailed observational perspective on the vertical cloud

and precipitation structure of WCBs during their inflow, ascent and outflow, and to evaluate their representation in the new ERA5 reanalysis dataset. To this end, satellite observations from the CloudSat radar and the CALIPSO lidar are combined with an ERA5-based WCB climatology for nine Northern Hemisphere winters. Based on a case study and a composite analysis, the main findings can be summarised as follows: (i) WCB air masses are part of deep, strongly precipitating clouds, with cloud-top heights at 9-10 km during their ascent, and an about 2-3 km deep layer with supercooled liquid water co-existing

with ice above the melting layer. The maximum surface precipitation occurs when the WCB is at about 2-4 km height. (ii) Convective clouds can be observed above the inflow and during the ascent. (iii) At upper levels, the WCB outflow is typically located near the top of a 3 km deep cirrus layer. (iv) There is a large variability between WCBs in terms of cloud structure, peak reflectivity, and associated surface precipitation. (v) The WCB trajectories with the highest radar reflectivities are mainly located over the North Atlantic and North Pacific, and – apart from the inflow – they occur at relatively low latitudes. They

are associated with particularly deep and strongly precipitating clouds that occur not only during the ascent, but also in the inflow and outflow regions. (vi) ERA5 represents the WCB clouds remarkably well in terms of position, thermodynamic phase and frozen hydrometeor distribution, although it underestimates the high ice and snow values in the mixed-phase clouds near the melting layer. (vii) In the lower troposphere, high potential vorticity is diabatically produced along the WCB in areas with high reflectivities and hydrometeor contents, and at upper levels, low potential vorticity prevails in the cirrus layer in the WCB

outflow. The study provides important observational insight into the internal cloud structure of WCBs, and emphasises the ability of ERA5 to essentially capture the observed pattern, but also reveals many small- and mesoscale structures observed by the remote sensing instruments but not captured by ERA5.

## 1 Introduction

Extratropical cyclones are typically associated with well-defined moist ascending airstreams referred to as warm conveyor belts

(WCBs; e.g., Harrold, 1973; Carlson, 1980; Wernli and Davies, 1997). WCBs are responsible for most of the cloud and pre-





cipitation formation and poleward energy transport in extratropical cyclones (Browning, 1990), and thereby they play a crucial role for the hydrological cycle and the Earth's energy balance. WCBs are also essential from an atmospheric dynamical point of view. The intense cloud-diabatic processes within the ascending airstreams lead to potential vorticity (PV) modifications in the lower and upper troposphere, which can intensify the associated cyclone (Binder et al., 2016) and influence the synoptic-

and large-scale flow at the tropopause and the downstream weather evolution (e.g., Wernli and Davies, 1997; Grams et al., 2011).

Because of the crucial role of WCBs for many atmospheric processes, it is important to accurately represent them and the associated clouds and precipitation in numerical weather prediction and climate models. Several studies indicated that errors in their representation can lead to forecast errors of the weather downstream (e.g., Gray et al., 2014; Martínez-Alvarado and

Plant, 2014; Madonna et al., 2015; Grams et al., 2018). For instance, it has been shown that cloud-microphysical processes in WCBs (Joos and Wernli, 2012; Joos and Forbes, 2016) and the initial moisture distribution in the WCB inflow (Schäfler et al., 2011; Schäfler and Harnisch, 2015) play a crucial role for the meso- and large-scale flow evolution. However, diabatic processes are difficult to represent in global models because they typically occur on smaller scales than the model resolution and must therefore be parameterised. Additionally, the understanding of many physical processes occurring in warm-, ice- and

particularly mixed-phase clouds is still incomplete (e.g., Illingworth et al., 2007; Joos and Forbes, 2016). This highlights the importance to complement our knowledge about WCBs and the embedded small-scale processes, which is mainly based on numerical model data, with observational studies.

Only a few observational studies exist on WCBs. Schäfler et al. (2011) compared lidar humidity observations of a summertime WCB over southwestern Europe with analysis data from the European Centre for Medium-Range Weather Forecasts

(ECMWF) and revealed significant deficiencies in the model's representation of the low-level moisture in the WCB inflow region. Crespo and Posselt (2016) analysed a WCB over the North Atlantic that was sampled several times by remote sensing instrumentation. They documented a clear transition from a stratiform to a convective cloud structure during the evolution of the cyclone. The distinction between mesoscale, slantwise ascent of the WCB and upright embedded convection had already been made in 1993 in a study on the mesoscale frontal structure of an explosively intensifying cyclone measured during the ERICA

field experiment (Neiman et al., 1993). Embedded deep convection was also documented in a number of WCBs observed in the Mediterranean region (Flaounas et al., 2016, 2018), and during the NAWDEX field experiment in the North Atlantic (Oertel et al., 2019). While slantwise WCB ascent leads to large-scale stratiform precipitation and the formation of widespread regions with low-PV air at upper levels, convective WCB ascent goes along with peaks of particularly strong surface precipitation and the formation of mesoscale upper-level PV dipoles, including regions with negative PV (Oertel et al., 2020). Finally, Gehring

et al. (2020) investigated the snowfall microphysical processes in a strongly precipitating wintertime WCB over the Korean Peninsula that was observed with radar data, radio soundings and snowflake photographs. They showed how the WCB created ideal conditions for rapid precipitation growth, including the formation of supercooled liquid water in the strongly ascending air masses, which favoured intense riming and aggregation.

Many cloud processes, for example radiative heating or cooling of the atmosphere, condensational processes and the ef-

ficiency of precipitation production, crucially depend on the vertical structure and distribution of clouds (e.g. Posselt et al.,



2008). With the launch of the CloudSat radar (Stephens et al., 2002, 2008) and the CALIPSO[1] lidar (Winker et al., 2003, 2009) in April 2006, high-resolution global measurements of the vertical structure and properties of clouds, precipitation and aerosols have become available. The satellites are part of the NASA's Afternoon Train (A-Train), a constellation of satellites that travels in close formation in a sun-synchronous orbit, allowing for near-simultaneous measurements of various key parameters that af-
fect the Earth's weather and climate. Studies based on CloudSat and CALIPSO measurements have provided invaluable insight into the distribution of clouds and precipitation in extratropical cyclones, and the associated complex dynamical and physical processes. Posselt et al. (2008) compared the frontal clouds observed by CloudSat along a warm, a cold and an occluded front, respectively, with those described in the Norwegian polar-front model (Bjerknes and Solberg, 1922). While the historical description and the modern observations reveal remarkable similarities, CloudSat provides a detailed view of the internal cloud
structure, thereby adding a new component to the classical conceptual model. Vertical composites of frontal clouds based on CloudSat and CALIPSO data (Naud et al., 2010, 2012, 2014, 2015) also revealed some similarities to the historical model, but even more to the conveyor belt model and specifically the WCB, with mid- and high-level clouds typically occurring to the east of the cold front and above the warm front. Field et al. (2011) combined observations from CloudSat and passive sensors to create three-dimensional composites of the cloud and precipitation structure in extratropical cyclones, and also used these
to evaluate the representation of clouds and precipitation in a numerical model. CloudSat and CALIPSO observations have also been used to validate the global ice cloud representation in several versions of the ECMWF and UK Met Office models with different ice cloud parameterisations (Delanoë et al., 2011). It was found that the models generally reproduce the main geographical and temperature-dependent distributions, although with some deficiencies, and that the representation is considerably improved in schemes with prognostic variables for cloud ice, snow, liquid water and rain compared to schemes with
diagnostic formulations for precipitation and mixed-phase clouds.

The objective of this study is to gain an observational view on a large number of WCBs in Northern Hemisphere winter, and to evaluate their representation in the new ERA5 reanalyses. For this, we combine satellite observations from CloudSat and CALIPSO with a WCB climatology compiled with ERA5. Specifically, the aims are to (i) use observational data to characterise the vertical cloud and precipitation structure of WCBs during their inflow, ascent and outflow in terms of vertical extent, radar
reflectivity and ice water content, (ii) gain insight into the ERA5-based meteorological environment of the observed WCBs in terms of saturation, static stability and PV, (iii) quantify differences in the cloud structure and the meteorological environment between typical WCBs and WCBs with particularly high radar reflectivities, and (iv) assess the ability of ERA5 to represent ice and snow in WCB clouds in comparison to CloudSat and CALIPSO.

The remainder of the paper is organised as follows. Section 2 describes the satellite measurements and ERA5 reanalyses,
as well as the method to combine the two datasets. In Section 3 we look at the cloud structure of an exemplary WCB in the North Pacific that was measured by the A-Train at the time of strongest intensity of the associated cyclone. The climatological analysis of the vertical cloud structure of WCBs is presented in Section 4, and a summary and discussion of the results are provided in Section 5.

---

[1]Cloud-Aerosol Lidar and Infrared Pathfinder Satellite Observation





## 2 Data and Methods

To characterise the vertical cloud and precipitation structure of the WCBs, and to gain insight into the meteorological environment they are embedded in, satellite observations from the CloudSat radar and the CALIPSO lidar are combined with ERA5-reanalyses from the ECMWF. The period chosen for the study extends from December 2006 to the end of January 2016, and the analysis is confined to Northern Hemisphere winter (December-February). Winter 2011/2012 is excluded from the study, as CloudSat was not part of the A-Train during that time period.

### 2.1    Satellite observations

Reflectivity profiles from the Cloud Profiling Radar (CPR) onboard the polar-orbiting CloudSat satellite (Stephens et al., 2002, 2008; Tanelli et al., 2008) are used. The CPR is a nadir-pointing radar operating with a frequency of 94 GHz ($\sim$ 3 mm, W-band). It provides cloud information with a vertical resolution of 485 m (oversampled to produce an effective resolution of 240 m) between the surface and 30 km altitude. The horizontal resolution is about 1.7 km in along-track direction, and 1.4 km in cross-
track direction. In the present study, we use CloudSat reflectivity data provided by the raDAR/liDAR (DARDAR) project (Delanoë and Hogan, 2010, see below), which is interpolated to a grid with 1.1 km horizontal and 60 m vertical resolution. The sensitivity of the CPR ranges from $-30$ to $50$ dBZ. Absorption by gases, liquid water and precipitating particles results in a two-way attenuation of the radar signal, which can amount to more than $10$ dBZ km$^{-1}$ with high liquid water contents and even a full attenuation of the radar signal in strongly precipitating systems (Mace et al., 2007; Marchand et al., 2008). Reflectivity
values between about $-30$ dBZ and $-15$ dBZ typically represent non-precipitating clouds, values between $-15$ dBZ and $0$ dBZ drizzle or light rain, and values greater than $0$ dBZ rain with increasing intensity (Stephens and Haynes, 2007; Haynes et al., 2009). According to Haynes et al. (2009), unattenuated near-surface reflectivity values of more than $0$ dBZ ($-5$ dBZ) are almost certainly associated with significant surface rain (snow) rates of at least $0.03$ mm h$^{-1}$. Radar signals likely contaminated by surface or clear air clutter are filtered out in the present analysis with the CloudSat cloud mask from the 2B-GEOPROF product
(Marchand et al., 2008). The effect of surface clutter is most pronounced below 1.2 km height (Marchand et al., 2008), which reduces the ability to investigate the cloud structure at the height of the WCB inflow.

Ice water content (IWC) profiles are obtained from the DARDAR cloud product, version v1 (Delanoë and Hogan, 2010). They are derived using a variational method that combines CloudSat radar reflectivities, CALIPSO lidar attenuated backscatter and infrared radiometer data of the Moderate Resolution Imaging Spectroradiometer (MODIS) on-board the Aqua satellite
(for details see Delanoë and Hogan, 2008, 2010). CloudSat and CALIPSO products are highly complementary and sensitive to very different powers of particle size. The radar is able to measure and penetrate optically thick clouds, but it cannot detect optically thin clouds with a reflectivity value below the minimum detectable signal of the radar ($-30$ dBZ). On the other hand, the lidar is sensitive to optically thin clouds, but it is strongly attenuated in optically thick clouds. Therefore, by linking CloudSat, CALIPSO and other A-Train measurements, DARDAR combines the advantages of the different sensors, thereby
providing a more detailed picture of the structure and microphysical properties of the hydrometeors than could be obtained by the individual sensors. The IWC retrievals consist of the entire frozen hydrometeor fraction, without distinction between





small ice crystals and sedimenting snow flakes. Uncertainties in IWC are estimated to reach up to about 60%. Despite these significant uncertainties, the DARDAR cloud product currently comprises one of the most accurate estimate of ice clouds properties (Stein et al., 2011; Eliasson et al., 2013). Like the reflectivity profiles, the IWC data is available on a grid with
1.1 km horizontal and 60 m vertical resolution.

## 2.2   ERA5 reanalyses

ERA5 reanalyses from the ECMWF (Hersbach et al., 2019) are used for the WCB identification and the characterisation of the meteorological environment. ERA5 is based on the Integrated Forecast System model version cycle 41r2 that was operational in 2016. The reanalyses have a spectral resolution of T639 (corresponding to ∼31 km) on 137 vertical levels and a temporal
resolution of 1 h. In this study, the fields are interpolated to a regular grid with $0.5°$ horizontal resolution.

In ERA5, the parameterization of stratiform clouds and large-scale precipitation is based on an advanced version of the scheme developed by Tiedtke (1993) and includes prognostic variables for water vapour, cloud liquid water, cloud ice, rain, snow and grid box fractional cloud cover (ECMWF, 2016; see also Forbes and Ahlgrimm, 2014). Interactions between the various water species are described with parameterisations for condensation, deposition and freezing via stratiform and convec-
tive processes, evaporation, sublimation and melting, as well as the generation of precipitation via autoconversion, accretion and snow riming. Precipitating particles have a terminal fall speed and can be advected between grid boxes by the three-dimensional wind. Supercooled liquid water can exist at temperatures between $0°C$ and the homogeneous freezing threshold at $-38°C$. When ice and supercooled liquid co-exist, the ice crystals grow at the expense of the liquid water droplets via the Wegener-Bergeron-Findeisen mechanism. Convection is parameterised by a bulk mass flux scheme based on Tiedtke (1989),
with a modified convective available potential energy closure (Bechtold et al., 2014). It considers deep, shallow and midlevel convection. The collective behaviour of a range of cumulus clouds in a grid cell is represented by a single pair of plumes describing the updraft and downdraft mass fluxes associated with the cloud ensemble, including the processes of entrainment of environmental air into the cloud, and detrainment of cloud condensate into the environment.

## 2.3   WCB identification

The ERA5-based WCB trajectories are calculated with a slightly modified version of the algorithm developed by Madonna et al. (2014). Based on the Lagrangian Analysis Tool (LAGRANTO; Wernli and Davies, 1997; Sprenger and Wernli, 2015), trajectories are started every 6 h from an equidistant grid in the lower troposphere and calculated forward for 48 h. For the climatological analysis, the starting points are located every 80 km in the horizontal direction and vertically every 20 hPa between 1050 and 790 hPa, consistent with Madonna et al. (2014). For the case study, the horizontal resolution of the starting
grid is increased to 40 km. To be classified as WCB air parcels, the trajectories must experience a strong ascent of at least 600 hPa within 48 h in the vicinity of an extratropical cyclone, whereby extratropical cyclones are identified as two-dimensional objects based on the algorithm of Wernli and Schwierz (2006), refined in Sprenger et al. (2017). To exclude rapid ascent associated with tropical cyclones, the WCB trajectories are required to be north of $25°N$ during their ascent phase (at time 24 h). With respect to the WCB identification method by Madonna et al. (2014), two modifications are made: (i) Very fast



ascending trajectories that fulfil the 600-hPa ascent criterion in the first part of the 48-h period and thereafter descend again are also selected. For those trajectories the pressure difference between times 0 and 48 h can be lower than 600 hPa and therefore they would have been neglected as WCB trajectories by the approach of Madonna et al. (2014). Compared to the original method, this leads to an increase in the number of identified WCB trajectories by about 35% (K. Heitmann, personal communication). (ii) WCB trajectories with a small distance to each other are clustered and considered to belong to the same

WCB, with a clustering method similar to the one described in Catto et al. (2015). When one of the trajectories in the cluster is collocated with a surface cyclone for at least one time step during the ascent, all the trajectories in the cluster are considered as WCB trajectories. Compared to the original algorithm, where every single trajectory is required to be collocated with a surface cyclone, this leads to a further increase in the number of WCB trajectories by about 10%. In the present study, the WCB trajectories are classified according to their height into inflow (0-2 km height; corresponding to about 1000-800 hPa),

ascent (2-7 km height and pressure $\sim$ 800-400 hPa) and outflow ($>$7 km height and pressure $<$ 400 hPa), respectively.

### 2.4 Matches between WCB trajectories and CloudSat – CALIPSO overpasses

To combine the ERA5-based WCB trajectories with the satellite data, all WCB trajectories are selected that are overpassed by the CloudSat – CALIPSO satellites. Since the satellite tracks are available at much higher temporal resolution than the hourly ERA5 data, the ERA5 fields are assumed to be representative for a 1-h time range of $\pm$ 30 minutes around each full hour.

A match between a WCB trajectory position and the satellites occurs when the trajectory is located within 50 km horizontal distance of the satellite orbit during this 1-hour window. In total, 502'977 matches are identified between individual WCB trajectories and A-Train overpasses in the nine winters. These matches are associated with about 9'000 different WCB clusters.

In the climatological study, several satellite profiles are attributed to each matching WCB trajectory – that is, the profile with the closest distance to the WCB air parcel, plus the 56 preceding and succeeding profiles. The average over the 113

assigned satellite profiles corresponds to a track segment length of about 124 km and therefore approximately the effective horizontal resolution of ERA5 (four times the horizontal grid spacing), i.e., the smallest scale the model is able to resolve fully (see Abdalla et al., 2013). This allows for a better comparison between the observations and the model data. Consistent with earlier work (Illingworth et al., 2007; Delanoë et al., 2011), it is assumed that the narrow satellite track is representative for the three-dimensional model grid box.

### 2.5 Selection of strong WCBs

In addition to the analysis of the entire climatology of matching WCB trajectories, those with highest reflectivity values given by CloudSat are investigated separately. More specifically, from all 502'977 matches, in each 0.5 km height bin the 5% with highest reflectivities at their respective height are selected and referred to as "strong WCBs". These top 5% in terms of reflectivity are assumed to be the strongest cloud-and-precipitation producing WCB air parcels. Comparison of the entire

climatology and the subcategory of "strong WCBs" allows to assess differences between the two categories in the cloud and precipitation structure, the geographical distribution, and the meteorological environment.



## 3 Case study of a representative North Pacific WCB

In this Section we examine the cloud and precipitation structure of a representative WCB that occurred in January 2014 over the central North Pacific. The associated cyclone underwent an explosive deepening and was observed by the A-Train at the time
of its strongest intensity (minimum sea level pressure of 975 hPa), around 00 UTC 3 January 2014. At this time the infrared satellite image and the overlaid ERA5 fields in Fig. 1a reveal a comma cloud pattern with high clouds along the cold, the warm and the intense bent-back front, and a distinct dry slot that wraps around the storm centre below a cyclonically breaking upper-level wave, shown by the 2-pvu contour on 315 K. The yellow contour outlines the grid points where, according to the reanalysis data, at least one WCB air parcel is present somewhere in the vertical column. The large area encompassed by this
contour indicates that the entire frontal region is influenced by WCB air. Note that these air parcels belong to WCB trajectories with a range of different starting times and vertical positions, with some still located at low levels at the beginning of their two-day ascent, some in the middle of their ascent, and some already located in the upper troposphere at the end of their two-day ascent. As an example, Fig. 1b shows WCB trajectories with starting times at 06 UTC 2 January, and their position 18 h later (black dots), at the time of the satellite overpass.

The A-Train moved from the southeast to the northwest over the warm sector, the cold and the warm fronts of the mature cyclone (blue line in Fig. 1a,b), and thereby simultaneously captured parts of the WCB inflow, ascent and outflow. In the warm sector, south of 27°N, the reflectivity profile measured by CloudSat (Fig. 2a,b) reveals shallow-to-midlevel convection above the WCB inflow, which is corroborated by the negative vertical gradient in equivalent potential temperature ($\theta_e$) between the surface and 4 km height. Between 27° and 33°N the WCB inflow is cloud-free and located below thin cirrus clouds in the
WCB outflow that increase in thickness toward the cold front. At the cold front, between 33° and 40°N, and above the surface warm front, between 42° and 46°N, most of the deep cloud system is associated with WCB air, in particular at the cold front (see purple dots in Fig. 2b that mark WCB air parcels). The high reflectivities (Fig. 2a) and satellite-retrieved IWC values (Fig. 2c) below 6-8 km indicate strong precipitation in the form of snow above and rain or melting snow below the 0° isotherm. Above 6-8 km, at temperatures colder than about −20°C, the lower reflectivity and IWC values indicate ice clouds rather than
falling snow. At the cold front, in the unstable air south of 37°N the presence of some narrow columns with particularly high reflectivities suggest convective WCB ascent embedded in the frontal cloud. In the northern part of the cold front, the higher static stability and horizontally relatively uniform reflectivities point to a mainly stratiform cloud structure. At the warm frontal zone the WCB intersections with the satellite track indicate a gentle slantwise ascent along the tilted moist isentropes (Fig. 2b). North of 46°N the associated ice clouds decrease in thickness with increasing distance from the surface warm front, as the
cloud base slopes upward along the moist isentropes and the cloud-top height decreases.

Figure 2d shows in colours the sum of the prognostic cloud ice and snow variables of ERA5, interpolated along the satellite track. To allow for a better comparison with the observations, in Fig. 2e the satellite-retrieved IWC is shown as the running mean over 113 satellite profiles, which corresponds to a track length of 124 km and thereby approximately the effective resolution of ERA5 (i.e., four times the horizontal grid spacing, see Abdalla et al., 2013). ERA5 captures the location of the cloud system and
the broad ice and snow structure remarkably well. The values strongly increase toward the melting layer and peak right above





the melting layer in the deep frontal clouds, where most of the frozen fraction is present as falling snow (see grey contours). However, the model considerably underestimates the peak values between the melting layer and the $-20°C$ isotherm (maxima of $1050\,\mathrm{mg\,m^{-3}}$ in ERA5 compared to $1630\,\mathrm{mg\,m^{-3}}$ in the observations), which are most likely associated with mixed-phase clouds. The underestimation is particularly pronounced in the convective clouds south of $27°N$. Furthermore, along the entire

cross section the cloud edges are less sharp than in the observations, and the transition between cloudy and cloud-free regions is smoother. Nevertheless, comparison with ERA5's predecessor, ERA-Interim, reveals a strong improvement in the ice cloud representation in ERA5 (Binder, 2016). In ERA-Interim, the agreement with the observed IWC is very poor, in particular in the mixed-phase clouds, where the underestimation of the high values close to the melting layer amounts to several orders of magnitude. The significant improvement of the representation from ERA-Interim to ERA5 can mainly be explained by a major

upgrade of the cloud and precipitation parameterisation (see also Delanoë et al., 2011; Forbes and Ahlgrimm, 2014): While ERA5 is based on prognostic variables for cloud ice, snow, liquid water and rain, in ERA-Interim precipitation and mixed-phase clouds are described by diagnostic formulations, and snow is not present in the IWC variable but directly removed from the atmospheric column.

In summary, the case study of this North Pacific WCB reveals that (i) WCB air parcels form part of vertically extended,

strongly precipitating clouds, but not the entire frontal cloud system is WCB air, (ii) convection can occur in the WCB inflow and ascent region, consistent with previous studies (e.g., Crespo and Posselt, 2016; Oertel et al., 2019), and (iii) the ERA5 reanalysis with prognostic variables for snow and ice is able to capture the broad structure and distribution of the frozen hydrometeor fraction associated with the WCB cloud, but the peak values are underestimated.

## 4 Climatological analysis of the WCB cloud structure

In this Section the WCB clouds and their meteorological environment will be characterised climatologically for nine Northern Hemisphere winters. We will first discuss the spatial distribution of the matches between WCB trajectories and the A-Train, and then investigate their vertical structure in satellite observations and reanalysis fields.

### 4.1 Spatial distribution of the intersected WCB trajectories

Figure 3a shows the spatial distribution of the 502'977 matches between individual WCB trajectories and the A-Train over-

passes. Matches occurred almost in the entire extratropics, but the highest number is present over the North Pacific and North Atlantic storm track regions between about $30°$ and $60°$. This spatial pattern reflects the winter climatological distribution of WCBs during their inflow, ascent and outflow (blue contours). In general, in the southwestern ocean basins most WCB air parcels were observed in the inflow, i.e., when the WCB trajectories were below $2\,\mathrm{km}$ height (pressure levels $> 800\,\mathrm{hPa}$), in agreement with the climatological maximum of WCB starting positions (Madonna et al., 2014). In contrast, in the northeastern

part of the oceans, at the end of the storm tracks, the majority of the matches occurred in the outflow, i.e., when WCB trajectories were above $7\,\mathrm{km}$ height (pressure levels $< 400\,\mathrm{hPa}$). Overall, the matches are distributed over a wide altitude range between the surface and $13\,\mathrm{km}$ height, with the highest numbers below $2\,\mathrm{km}$ in the inflow and in particular between 7 and





10 km in the midlatitude outflow (Fig. 3c). The latitude of the matches increases from about 25°-50°N in the WCB inflow to 35°-75°N for air parcels at 8 km height (mean and 10-90 interpercentile range in black in Fig. 3d), reflecting the poleward mo-
tion typically occurring during the WCB ascent. Note, however, that the WCB air parcels in the different height bins generally belong to different WCB trajectories – only in rare cases the same trajectory has been observed more than once by the narrow CloudSat–CALIPSO tracks. Matches above 8 km typically occurred again at lower latitudes (where the tropopause is higher than in polar region), ranging from about 30° to 60°N for air parcels at 10 km height, and from 10° to 35°N for very high outflow above 12 km. These matches are most likely associated with WCBs in convectively active subtropical systems.
The matches with "strong WCBs", that is, the 5% of the air parcels in each 0.5 km height bin with the highest CloudSat radar reflectivities (see Section 2.5), are mainly located over the North Pacific and North Atlantic (Fig. 3b). Interestingly, in the inflow and early ascent ($< 3.5$ km) the mean latitude of the WCB air parcels with exceptionally strong reflectivities is further north than the mean over all trajectories, with most air parcels located between 30° and 60°N (mean and 10-90 interpercentile range in red in Fig. 3d). The opposite is true at higher altitudes, where the strong WCBs are typically located much further
south than the entire climatology. Between 4 and 10 km height their mean latitude is approximately constant around 38°N, and matches at higher levels occur again at very low latitudes and are probably related to subtropical cyclones. Note, again, that different WCBs contribute to the strong category in each height bin, and the general decrease in the mean latitude from inflow to outflow observed for the strong WCBs does not imply an equatorward ascent. All in all, the analysis shows that WCBs with exceptionally strong radar reflectivities in their inflow occur further north than the mean of all matches, while higher than usual
radar reflectivities in the WCB ascent and outflow are found further south.

## 4.2 Composites of reflectivity and DARDAR-retrieved IWC

To investigate the cloud structure of the WCBs during their inflow, ascent and outflow, we create vertical composites of the satellite observations, separately for different WCB heights. Hereby, all matching WCB air parcels are classified into 0.5 km height bins, with the number of matches per bin shown in the histogram in Fig. 3c. We will first examine the composites
associated with the entire climatology, and then compare them to the subcategory of strong WCBs.

### 4.2.1 All WCBs

Figure 4 shows composites of vertical profiles of CloudSat reflectivity (Fig. 4a) and DARDAR-retrieved IWC (Fig. 4b) as a function of the height at which the A-Train profile matched with a WCB trajectory. This height is referred to as $z_{WCB}$. For instance, for matches at a height of $z_{WCB} = 3$ km, the radar reflectivity shows median values[2] exceeding $-6$ dBZ from
near the ground to about 6 km altitude, whereas for matches above $z_{WCB} = 7$ km, radar reflectivities are below $-30$ dBZ, if the median is calculated over all WCBs (Fig. 4a). Thus, with increasing $z_{WCB}$ the composites give insight into the vertical cloud structure associated with the WCB inflow ($z_{WCB} = 0\text{-}2$ km), ascent ($z_{WCB} = 2\text{-}7$ km) and outflow ($z_{WCB} > 7$ km), respectively. Comparison with the latitude-height distribution of the matches (thick black line in Fig. 3d) shows that up to

---

[2]For the reflectivity and IWC fields it is more meaningful to show the median value rather than the mean. In the case of reflectivity this allows us to take into account clear-sky values, and in the case of IWC the mean would be dominated by large values.





$z_{WCB} \approx 8$ km the satellite composites can be interpreted as a – somewhat irregular – vertical cross section from south to
north through poleward ascending WCB air. However, keep in mind that the matching WCB air parcels in the different height
bins correspond to different WCB trajectories, such that Fig. 4 shows a composition of single WCB positions and not the
development along individual trajectories. As seen in the case study, several matches can occur on top of each other in the
same cloud system (see purple dots in Fig. 2b). In these cases, in the composites the same satellite profile is taken into account
for each match separately, i.e., it is present more than once along the x-axis. In summary, this sophisticated compositing
approach is able to provide representative vertical profiles of observed radar reflectivity and IWC along WCBs in the Northern
Hemisphere storm track regions in winter.

To quantify how often WCB matches occur on top of each other in the same cloud system, the contours in Fig. 4a,b show
the relative WCB trajectory frequency, which is the number of matches in a certain 0.5 km profile height bin normalised by the
total number of matches in the specific WCB height bin. For instance, a relative frequency of 25% at $z_{WCB} = 3.75$ km and a
profile height $z_{prof} = 8.75$ km indicates that 25% of the WCB trajectories located between 3.5 and 4 km height have another
WCB trajectory on top of them between 8.5 and 9 km height. Such a situation occurred, for instance, in the case study between
about 36° and 45°N in the deep cold and warm frontal clouds (Fig. 2b). For matches at $z_{WCB} = 1$ km height, only in 10% of
the cases another WCB trajectory is located on top of them at 8.5-9 km height. The vertical area in between can either also be
associated with WCBs (as in the case study between 36° and 42°N; Fig. 2b) or not (as between 30° and 36°N in Fig. 2b) –
the frequency values do not provide insight into the vertical connectivity of the WCB air. By definition, the frequency is 100%
along the diagonal line, and the slower its decrease away from the diagonal the more the observed reflectivity and IWC patterns
are associated with WCB air.

In the reflectivity composite, the white areas indicate either clear air, or, in particular below about 0.8 km height, ground
clutter, which both have been filtered out (see Section 2.1). In the IWC composite, the white areas indicate the absence of
frozen hydrometeors. The cloud-top heights according to the radar reflectivities are everywhere lower than those according
to the IWC retrievals. This can be explained by the inability of the radar to detect thin ice clouds with reflectivities below
$-30$ dBZ, whereas for the IWC retrievals the radar measurements are combined with lidar data that is strongly sensitive to
optically thin ice clouds (see Section 2.1).

Overlaid on top of the observed fields are ERA5-based temperature contours (red dashed lines in Fig. 4a,b) and the dynamical
tropopause (brown 2 pvu contour in Fig. 4a,b). From $z_{WCB} = 0$ km up to about $z_{WCB} = 8$ km (thick black line in Fig. 3d),
the melting layer and the dynamical tropopause decrease in height (Fig. 4a,b), consistent with the increase in mean latitude
(thick black line in Fig. 3d), while between $z_{WCB} = 8$ and 14 km the transition from predominantly subpolar to subtropical
outflow is associated with an increase in their heights. At the height of the WCB inflow ($z_{WCB}$ and $z_{prof} \approx 0\text{-}2$ km), liquid
clouds with reflectivities of about $-10$ to $-5$ dBZ are present above the ground clutter (Fig. 4a), indicating some drizzle or
light rain (Stephens and Haynes, 2007). Above the inflow, the small non-zero median IWC values reveal the presence of thin
ice clouds that extend up to about 9 km height (Fig. 4b). As suggested by the low reflectivities below $-30$ dBZ, this is most
likely non-precipitating cirrus. The 10% WCB trajectory contour approximately follows these ice clouds, indicating that a
small fraction of them is associated with WCB air.





During the ascent ($z_{WCB} \approx$ 2-7 km) WCBs form part of deep clouds, with cloud-top heights just below the ERA5-based
dynamical tropopause at 9-10 km (Fig. 4a,b). The clouds have relatively high reflectivity values up to $> 4\,$dBZ and – between
the melting layer and about $-20°$C – high DARDAR-retrieved IWC with peaks at $260\,$mg m$^{-3}$, which indicates precipitation
in the form of snow above and rain below the melting layer. The highest IWC values occur approximately at the height of the
WCB, whereas the highest reflectivities are present just below the WCB near the melting layer. The peak values, and therefore
most likely the strongest surface precipitation, occur when the WCB trajectories are at a height of about $z_{WCB} = 3.5$-$4\,$km.
Compared to the inflow, the WCB trajectory densities above and below the diagonal are higher in most of the ascent region,
indicating a stronger contribution of WCB air to the reflectivity and IWC profiles. Of course, there is a large case-to-case
variability, and the median profiles contain cloud systems that are almost entirely formed by strongly ascending WCB air, and
others where only a small part of the deep cloud is WCB air. The decrease in the WCB densities to only 10-25% at cloud base
and cloud top suggests that the latter is often the case, i.e., the WCB ascent is typically embedded in a deeper cloud that forms
partially due to air parcels with a weaker ascent than required for the WCB criterion.

In contrast to the ascent, the WCB outflow ($z_{WCB} > 7\,$km) is located within an about 3 km deep cirrus layer with low
reflectivities below the sensitivity of the radar (Fig. 4a) and low IWC (1-10 mg m$^{-3}$, Fig. 4b). Their cloud bases and tops
increase gradually with increasing outflow height, which suggests a predominantly slantwise ascent along the baroclinic zone,
similar to the pattern observed in the case study north of 46°N (Fig. 2). The deep cirrus layer and the extension of ice clouds
above the WCB outflow level are in line with the findings of Wernli et al. (2016), who showed that the WCB outflow is often
associated with cirrus clouds that form by freezing of liquid droplets during the strong ascent, while above the outflow in situ
ice cloud formation can occur in response to the strong lifting associated with the WCB.

### 4.2.2 Strong WCBs

Analogous to the satellite composites created for all WCB matches, Fig. 4c,d shows the median reflectivity and IWC profiles
for the subcategory of strong WCBs, together with ERA5-based temperature contours and the dynamical tropopause. The
slight decrease in latitude with increasing height of the WCB matches (red line in Fig. 3d) is reflected in a general increase
in the melting layer height and the tropopause along the x-axis (Fig. 4c,d), in contrast to the decrease observed for the entire
climatology (Fig. 4a,b). It is again important to keep in mind that these composites cannot be interpreted in a Lagrangian way;
the WCB air parcels in the different height bins generally belong to different WCB trajectories and the overall decrease in
latitude with increasing $z_{WCB}$ does not imply an equatorward ascent.

At the height of the WCB (along the diagonal), but also above and below, the reflectivities exceed those of all matches by
10-25 dBZ. Above the melting layer this goes along with DARDAR-retrieved IWC values that are a factor of 5-5000 larger
than those of all matches (Fig. 4d). This confirms that the top 5% of the matches in terms of radar signal are indeed very
strongly cloud-and-precipitation-producing WCB air trajectories. Deep clouds extending from the surface to the tropopause
occur not only in the ascent region, but – in contrast to all matches – also above the inflow and in the outflow. The cloud-top
height according to the IWC pattern is at 10 km above the air parcels in the WCB inflow, at 11 km for the ascending ones and at
11-14 km for the outflow, and it is thus everywhere higher than for the entire climatology. The decrease in the WCB trajectory





densities away from the diagonal is slower than for all matches, implying a stronger contribution of WCB trajectories to the formation of the deep clouds. Presumably, WCB trajectories in the inflow, ascent and outflow are often located on top of each

other in the same cloud system, as seen in the case study at the cold front (Fig. 2b). These results also indicate that a vertically deep layer of trajectories fulfilling the WCB criterion is likely leading to particularly high reflectivities and intense surface precipitation. The peak reflectivities (13.9 dBZ compared to 4.4 dBZ for all matches) occur at the height of the ascending WCB and not below as for all matches, and there is no clear indication for a melting layer in the radar signal. The IWC maxima (1190 mg m$^{-3}$ compared to 260 mg m$^{-3}$ for all matches) are collocated with the reflectivity maxima. They extend over several

WCB heights along the diagonal, from about $z_{WCB} = 2.5$ km to $z_{WCB} = 5$ km, in contrast to the rather localised peak at $z_{WCB} = 3.5$-4 km observed for all matches.

### 4.3 Meteorological environment in ERA5

To analyse the WCB clouds and the differences between all and strong WCBs in more detail, we complement the satellite observations with model data from ERA5. This allows us to gain insight into the meteorological environment associated with

the matches, and, at the same time, to compare the modelled ice and snow water content with the DARDAR observations. To this end, vertical profiles of the ERA5 fields are interpolated to the position of the matches, and composites equivalent to those discussed in the previous Section are created. Again, we will first examine the fields for all WCB matches and then compare them with the subcategory of strong WCBs.

#### 4.3.1 All WCBs

The vertical composites of various ERA5 fields are shown in Fig. 5, separately for different WCB heights.[3] The frozen hydrometeor fraction, i.e., the sum of the prognostic cloud ice and snow variables (Fig. 5a), resembles the observed pattern in Fig. 4b remarkably well. However, in the ascent region between the melting layer and about $-20°$C the peak values are underestimated (160 mg m$^{-3}$ in ERA5 vs. 260 mg m$^{-3}$ in the observations), whereas above the inflow and below the outflow the values are overestimated, and the cirrus layer associated with the outflow is considerably deeper than in the observations. Also

the transition between cloudy and cloud-free regions is smoother. The underestimation of the peak values close to the melting layer is consistent with the case study. It occurs in a 2-3 km deep layer with mixed-phase clouds (Fig. 5b), which are known to be difficult to simulate and associated with large uncertainties in many numerical weather prediction and climate models (e.g., Morrison et al., 2003; Illingworth et al., 2007; Klein et al., 2009; Delanoë et al., 2011).

The ice clouds in ERA5 coincide with high relative humidities with respect to ice (RH$_{ice}$; Fig. 5c). The values are close

to saturation ($> 80\%$) along most of the WCB on the diagonal, and they are particularly high in a deep layer in the ascent region, where the strong updraft continuously leads to new ice cloud formation. At lower altitudes, the inflow and especially the ascent regions are also associated with high cloud liquid water and rain water contents (Fig. 5b). The liquid hydrometeor fraction extends from the surface up to about 5-6 km height, with a 2-3 km deep layer of supercooled liquid water co-existing

---

[3]Except for the frozen and liquid hydrometeor content the mean rather than the median profiles are shown in each height bin, as the fields are slightly smoother, but the median profiles are very similar.





with ice above the melting layer. The highest liquid hydrometeor values occur during the WCB ascent ($z_{WCB}$ at $\approx$ 2-4 km)
slightly below the WCB and the melting layer, in the lower part of the vertically extended cloud. Accordingly, also the sur-
face precipitation has a maximum when the WCB is at 2-4 km height, with values of about $1.8 \, \text{mm h}^{-1}$ (solid green line in
Fig. 6). Surface precipitation is also high in the WCB inflow (1.2-1.5 mm h$^{-1}$), whereas it is rather weak below the outflow
(0.3 mm h$^{-1}$). For $z_{WCB} > 6 - 7$ km, the frozen and liquid hydrometeors are vertically disconnected (Fig. 5a,b), consistent
with the tongue of relatively low RH$_{ice} < 70\%$ in between (Fig. 5c). This suggests that the weak surface precipitation evident
for $z_{WCB} > 6 - 7$ km (Fig. 6) is not associated with the WCB, but with the low-level warm clouds present at 1-2 km height
below the WCB (Fig. 5b). Throughout the inflow, ascent and outflow, most of the surface precipitation is associated with the
large-scale cloud scheme (Fig. 6). Convective precipitation is significantly lower, but has a small peak in the inflow and early
ascent. The presence of convection in the inflow is further corroborated by very low moist static stability values (i.e., weak
vertical gradients in equivalent potential temperature, $d\theta_e/dz$) in that region (Fig. 5d,e). Higher stabilities are present below the
ascending WCB, where the tilted moist isentropes indicate a lifting over the cold or warm front (Fig. 5d). At higher altitudes
during the ascent, at and above the height of the WCB, the stabilities are again lower and indicate some convective motion.

The strong cloud and precipitation formation goes along with elevated PV values in the lower and middle troposphere, and
low PV in the WCB outflow (Fig. 5f). The elevated low and mid-level PV ($> 0.5$ pvu) extends over a broad and deep region
in the inflow, ascent and early outflow ($z_{WCB}$ at $\approx$ 1-10 km), and coincides with increased stability (Fig. 5e). Two areas with
particularly high PV ($> 0.7$ pvu) are located slightly below the ascending WCB ($z_{WCB}$ at $\approx$ 2-5 km; Fig. 5f). One of these
two high low-level PV areas is located between the melting layer and the observed and modelled snow and ice maximum at
WCB height (Figs. 4b and 5a), and it coincides with the radar reflectivity maximum (Fig. 4a). Most likely, latent cooling of the
melting layer, as well as latent heating due to freezing of cloud water and vapour deposition on ice particles along the WCB,
both contribute to the PV maximum in between. The agreement with the observations indicates that in addition to the good
representation of cloud ice and snow in ERA5, the reanalysis data is able to capture the cloud-diabatic processes associated
with WCBs and their impact on the dynamics very well. The second area with high low-level PV (at $z_{WCB} \approx$ 2-3 km) is located
below the melting layer at 1-2 km profile height and coincides with the maximum in the modelled cloud rain and liquid water
content (Fig. 4b), which suggests that here the PV production is mainly associated with latent heating due to condensation,
and potentially some below-cloud cooling due to rain evaporation. This PV maximum is not accompanied by a corresponding
maximum in reflectivity (Fig. 4a), probably as a result of the two-way attenuation of the radar signal close to the surface in
strongly precipitating systems. In the WCB outflow, the PV values are anomalously low ($< 0.2$ pvu; Fig. 5f) and coincide with
reduced vertical stability (Fig. 5e). As a consequence of the low-PV air in the outflow, the tropopause above is elevated, and a
sharp vertical PV gradient is established between the low values at WCB height and the high values in the stratosphere. The
elevated tropopause also goes along with an increased vertical gradient in equivalent potential temperature and a layer with
peak vertical stability in the stratospheric air above the WCB outflow (Fig. 5d,e), which is referred to as tropopause inversion
layer (TIL; Birner et al., 2002). As discussed by Kunkel et al. (2016), the TIL typically forms above the WCB, because (i) the
low-level cloud-diabatic processes lead to an increase in the vertical motion and an enhancement of static stability above the





updraft region, and (ii) the upward transport of moisture into the tropopause region and the formation of high-level ice clouds goes along with strong radiative cooling at the tropopause, which contributes to a further enhancement of the TIL.

### 4.3.2 Strong WCBs

The vertical composites of the ERA5 fields for the subcategory of strong WCBs are shown in Fig. 7. Again, the reanalysis correctly captures the broad structure and distribution of ice and snow along the WCBs (compare Fig. 7a and Fig. 4d), but underestimates the peak values in the mixed-phase layer (490 mg m$^{-3}$ in ERA5 vs. 1190 mg m$^{-3}$ in the observations). Consistent with the satellite measurements (Fig. 4c,d), throughout the inflow, ascent and outflow the modelled clouds associated with the strong WCBs are considerably deeper than those associated with the entire climatology, their hydrometeor contents are higher, the layer with mixed-phase clouds is deeper (Fig. 7a,b), and RH$_{ice}$ is higher (Fig. 7c). This goes along with considerably higher surface precipitation along the entire WCB, with peak values above 3 mm h$^{-1}$ when the air parcels are at 3.5 km height, and significant amounts also in the inflow and below the outflow (solid blue line in Fig. 6). In contrast to all WCB matches, up to $z_{WCB} \approx 9$ km there is no gap between the frozen and liquid hydrometeors (Fig. 7a,b), consistent with the higher RH$_{ice}$ (Fig. 7c). This suggests that also above $z_{WCB} \approx 6$ km the surface precipitation is coming from the WCB, and not from the low-level warm clouds as in the case of all matches. Along the entire WCB, most of the precipitation is again associated with the large-scale cloud scheme. The strongest convective precipitation occurs for WCBs around 4-5 km height and is potentially linked to a local minimum in moist stability in the lower troposphere in that region (Fig. 7d,e).

The stronger cloud and precipitation formation is reflected in stronger low-level PV production, in particular for WCBs in the inflow and early ascent, with peak values $> 0.9$ pvu (Fig. 7f). In contrast to the entire climatology, where the highest PV values are located just below the WCB, for the strong category they occur along the WCB on the diagonal and coincide with the observed reflectivity and ice and snow maxima (Fig. 4c,d). Along the WCB, we expect the strongest latent heating due to condensation, freezing and vapour deposition on ice, consistent with the relative humidity maximum (Fig. 7c). Thus, the positive PV maximum coincides with the latent heating maximum, which is in line with the findings from previous studies (e.g., Wernli and Davies, 1997). In the WCB outflow, the PV values are anomalously low PV ($< 0.2$ pvu). As for the entire climatology, above the outflow the tropopause is elevated and a tropopause inversion layer is evident (Fig. 7e,f).

### 5 Summary and discussion

In this study, ERA5 reanalyses have been combined with satellite observations from the polar-orbiting CloudSat radar and CALIPSO lidar to gain a detailed observational perspective on the vertical cloud structure of WCBs during their inflow, ascent and outflow, and to evaluate their representation in ERA5. To this end, more than 500'000 matches between the satellite observations and ERA5-based WCB trajectories (corresponding to about 9'000 different WCB clusters) were evaluated during nine Northern Hemisphere winters in a composite analysis and a detailed case study. The majority of the matches occurred over the ocean basins in regions of high climatological WCB frequencies and can therefore be considered as representative.





The satellite observations revealed that the WCBs form part of vertically extended, strongly precipitating clouds, in particular
during their ascent, with cloud-top heights at 9-10 km. In some cases the entire cloud system is associated with WCB air, but
often the cloud parts below and above the WCB air parcels form in air with a comparatively weak ascent below the WCB
threshold. Convection can occur above the WCB inflow and during the ascent, in agreement with recent studies on convection
embedded in WCBs (e.g., Crespo and Posselt, 2016; Flaounas et al., 2016; Oertel et al., 2019). In the upper troposphere, after
the main ascent phase, the WCBs are typically located near the top of an about 3 km deep layer with cirrus clouds.

According to ERA5, high low-level PV occurs below the ascending WCB in the lower part of the vertically extended
cloud. The strongest low-level PV production occurs at about 3 km height between the melting layer and the ascending WCB
and coincides with the radar reflectivity maximum. It is most likely produced by a combination of diabatic heating due to
freezing of cloud water and depositional growth of ice particles at the WCB height and diabatic cooling from snow melting
below. A second area with particularly strong low-level PV production occurs at about 1-2 km height, below the WCB and
the melting layer, in a region with strong cloud-condensational heating, and possibly some below-cloud evaporative cooling.
The occurrence of the strongest positive PV anomalies below rather than at the WCB height is surprising, and the potentially
important contribution of various in- and below-cloud microphysical processes to the low-level PV production is in line with
the findings from recent modelling studies (Joos and Wernli, 2012; Crezee et al., 2017; Attinger et al., 2019).

The WCB trajectories with the highest reflectivity values ("strong WCBs") have mainly been observed over the North At-
lantic and North Pacific, and – except in the inflow – at relatively low latitudes ($\sim 38°$N). They are associated with particularly
deep and strongly precipitating clouds that occur not only during the ascent, but also in the inflow and outflow region. Com-
pared to the climatology of all WCBs, the hydrometeor content is considerably higher and the surface precipitation is stronger.
The low-level PV production is larger and has its peak in the inflow and early ascent at the height of the WCB, and it coincides
with high reflectivities and hydrometeor values. The agreement of the positive PV anomaly with the region of strong cloud for-
mation is characteristic for strongly ascending WCB air masses, where the PV anomaly, which is produced below the diabatic
heating maximum, is advected upward toward the heating maximum (Wernli and Davies, 1997).

The comparison between the satellite retrievals and ERA5 showed that the reanalyses are able to capture the main structure
of the WCB clouds in terms of position and thermodynamic cloud phase. The spatial pattern of the frozen hydrometeor fraction
(ice and snow) is in good agreement with the observations, in particular at high altitudes, where most of the frozen fraction
is present as ice rather than falling snow. However, the peak values in the mixed-phase cloud regime near the melting layer
are underestimated by a factor of 1.6 for the entire climatology and a factor of 2.4 for the subcategory of strong WCBs.
This corroborates the findings from many other studies that mixed-phase clouds are difficult to simulate accurately (e.g.,
Illingworth et al., 2007; Delanoë et al., 2011). The co-existence of ice, supercooled liquid water and water vapour, and the
complex interaction of various microphysical processes, render their understanding and parameterization in numerical weather
prediction and climate models particularly challenging. Nevertheless, compared to the older reanalysis dataset ERA-Interim
(Dee et al., 2011), where the microphysical parameterisations are based on simplified diagnostic relationships for snow and
mixed-phase clouds, the improved scheme in ERA5 with prognostic variables for cloud ice, snow, liquid water and rain leads to
a much more realistic representation of the WCB cloud pattern (Binder, 2016). This is consistent with the findings from Delanoë



et al. (2011) and Forbes and Ahlgrimm (2014), who also compared CloudSat–CALIPSO observations with two ECMWF
models with schemes similar to the ones in ERA5 and ERA-Interim, respectively, and found a significant improvement in the
ice-cloud parameterization with the upgrade from the diagnostic to the prognostic representation of mixed-phase clouds and
precipitation.

Following earlier studies (e.g., Illingworth et al., 2007; Delanoë et al., 2011), in the present analysis it has been assumed that
the cloud structure observed along the narrow two-dimensional satellite track is representative for the entire three-dimensional
volume of the model grid box. Despite a quite good agreement between ERA5 and the observations, it is possible that this
assumption is not entirely justified and the clouds observed by the satellite are not representative for the larger-scale fea-
tures. As proposed by Delanoë et al. (2011), additional information on the large-scale environment could be obtained from
other instruments onboard the A-Train satellites with a wider swath width, in order to better assess the representativity of the
measurements.

The CloudSat and CALIPSO measurements have provided a much needed, detailed observational perspective into the in-
ternal cloud structure of WCBs, and have revealed many small- and mesoscale structures not resolved by the temporally and
spatially much coarser-resolution model data that has mainly been used so far to study WCBs. The measurements comple-
ment the insight gained on WCBs from recent modelling studies with high-resolution convection-permitting simulations (e.g.,
Oertel et al., 2020). In future work, the large number of WCB trajectories observed by CloudSat and CALIPSO could still be
exploited in more detail, both in case studies and climatological analyses. They could be classified, for instance, according to
different criteria like the position relative to the cyclone centre and the stage of the cyclone life cycle, the ascent behaviour
(slantwise vs. convective), the outflow curvature (cyclonically vs. anticyclonically), the amplitude of the low-level positive
and upper-level negative PV anomalies or the geographical region, in order to assess whether different types of WCBs have
common characteristics. It would also be insightful to extend the climatological analysis to different seasons and the South-
ern Hemisphere, to investigate potential seasonal and hemispheric differences in the WCB cloud structure. Finally, the study
could be repeated with data from other models. In particular, it would be interesting to evaluate the representation of WCBs in
operational numerical weather prediction models at different lead times, which might allow for the identification of systematic
forecast errors.

*Data availability.* The DARDAR satellite products used in this study can be accessed from the ICARE website (http://www.icare.univ-
lille1.fr), and the ERA5 reanalyses from the ECMWF website (https://www.ecmwf.int/en/forecasts/datasets/reanalysis-datasets/era5). The
matches between the WCB trajectories and the satellite data are available from the authors upon request.

*Author contributions.* M.S. calculated the ERA5 WCB climatology. H.B. performed all other analyses in this study and wrote the paper. All
co-authors contributed to the interpretation of the results and writing of the paper.



*Competing interests.* The authors declare that they have no conflict of interest.

*Acknowledgements.* H.B. acknowledges funding from the Swiss National Science Foundation (SNSF) via grants 146834 and 185049, and M.B. acknowledges funding from the SNSF via grant 146834 and the European Research Council 485 (ERC) under the European Union's Horizon 2020 research and innovation programme (project INTEXseas, grant agreement No. 787652). We thank the ICARE data and services center for providing access to the DARDAR satellite data, and MeteoSwiss and ECMWF for access to the ERA5 reanalyses. We are grateful to Julien Delanoë, Paul Field and Catherine Naud for valuable comments and discussions.




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

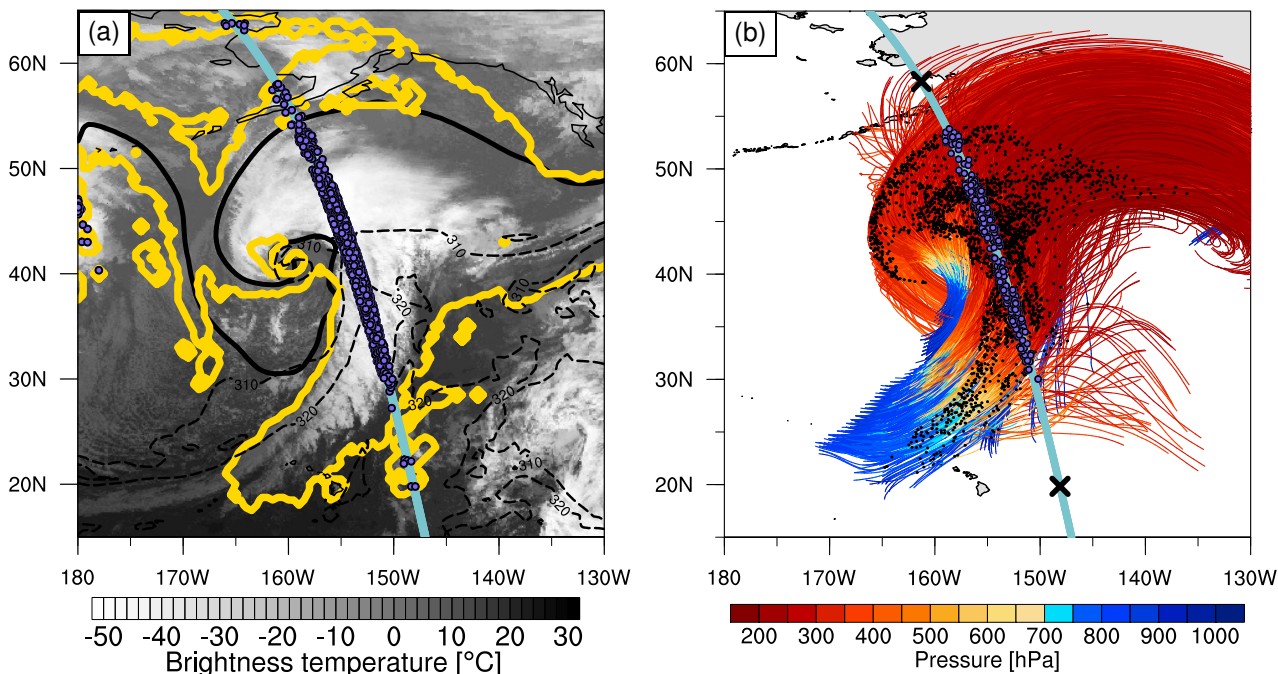

**Figure 1.** Case study of a North Pacific WCB at 00 UTC 3 January 2014. (a) Infrared satellite image (brightness temperature in °C) derived from the GridSat-B1 data (Knapp et al., 2011), and, from ERA5, the 2-pvu contour on 315 K (black), $\theta_e$ (black dashed contours at 310 and 320 K) and grid points with at least one WCB trajectory (yellow). The blue line marks the track of the A-Train, and the purple dots show matches between WCB air parcels and the satellite track. (b) Two-day WCB trajectories (coloured by pressure; hPa) starting at 06 UTC 2 Jan 2014. The positions of the trajectories at the time of the satellite overpass, at 00 UTC 3 Jan, are shown by black and purple dots, with the purple dots indicating matches with the satellite track (blue line). The blue line again marks the track of the A-Train, and the segment between the two black crosses indicates the region shown in Fig. 2.



**Figure 2.** Vertical cross sections of observed and modelled variables along a North Pacific WCB at 00 UTC 3 January 2014. (a) CloudSat radar reflectivity (dBZ; shading) along the segment between the two black crosses in Fig. 1b, together with, from ERA5, interpolated $\theta_e$ (black contours every 5 K), temperature (red dashed contours at $0°$, $-20°$ and $-40°$C) and the 2-pvu contour (thick black line). The labels mark the position of the warm sector ("WS"), the cold front ("CF") and the warm front ("WF"), respectively. (b) Same as (a), but with the positions of the intersected WCB trajectories shown by the purple dots. (c) DARDAR-retrieved IWC (mg m$^{-3}$; shading), (d) ERA5-based sum of IWC and SWC (mg m$^{-3}$; shading), and SWC only (thin black contours at 0.1,1,10,100,200 and 1000 mg m$^{-3}$), and (e) DARDAR-retrieved IWC in a running mean along 113 satellite profiles (corresponding to a segment of $\sim 124$ km). The thick black line in (c-e) again marks the 2-pvu contour, and the red dashed lines the $0°$, $-20°$ and $-40°$C isotherms.

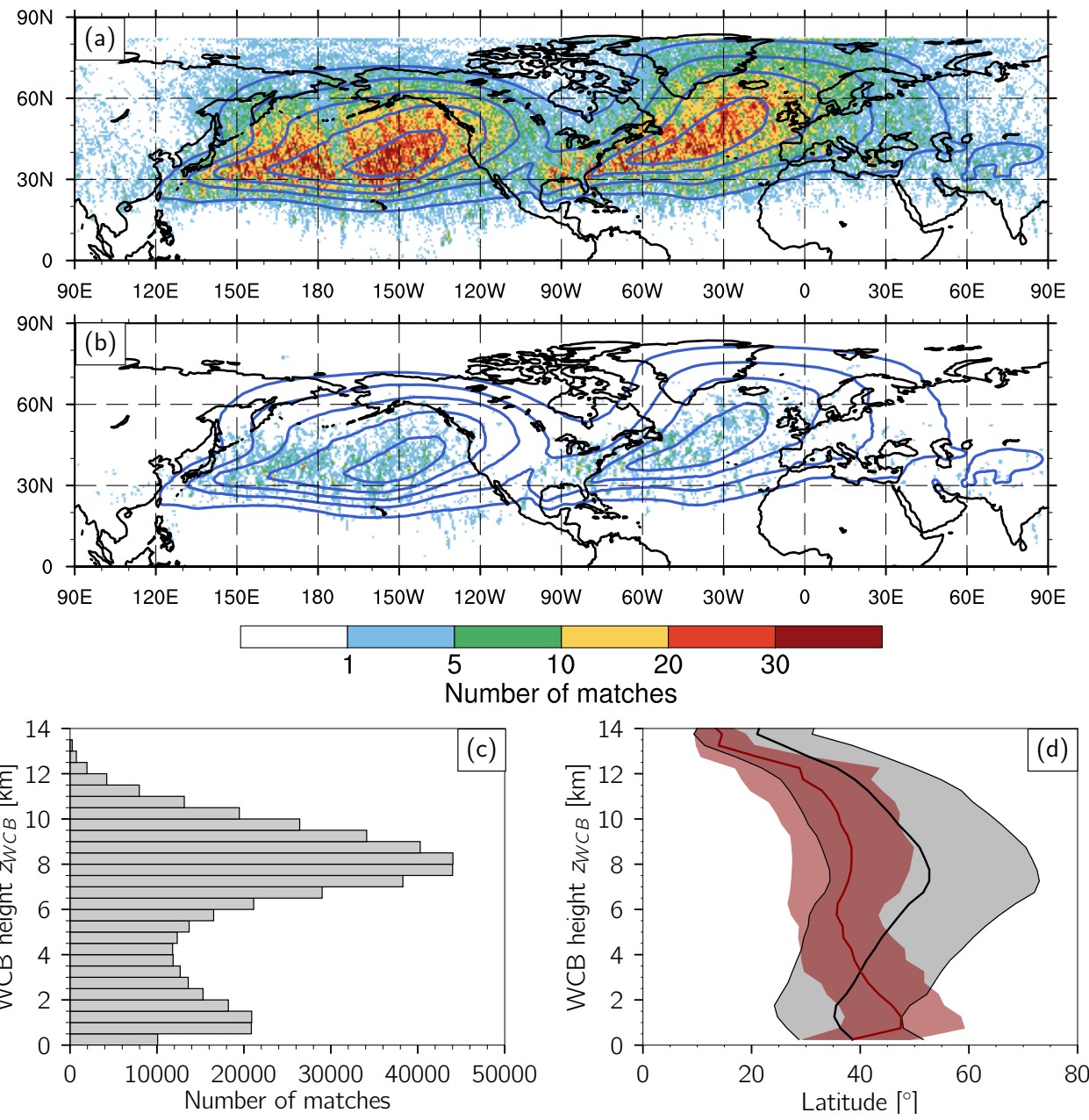

**Figure 3.** (a) Spatial distribution of the WCB trajectories matching with the satellite track (shading). The colours indicate the number of matches in each 0.5° bin. Overlaid is the ERA5-based climatological frequency of WCB trajectories for December-February 1980-2018 (blue contours every 10%), whereby all time steps between the start (t=0 h) and the end (t=48 h) of the trajectories are considered. (b) As in (a), but for strong WCBs, i.e., the top 5% of the matches with highest reflectivity values in each vertical height bin. (c) Height distribution of the matches, and (d) latitude of the matches as a function of their height for all WCBs (black and grey) and strong WCBs (red). The thick black and red lines show the mean over the matches, and the grey and red shading represent the range between the 10% and the 90% percentile.



**Figure 4.** Composites for all WCBs (top panels) and strong WCBs (bottom panels) of the median vertical profiles of (a, c) CloudSat radar reflectivity (dBZ; shading) and (b, d) DARDAR-retrieved IWC (mg m$^{-3}$; shading), separately for different height bins of the matching WCB air parcels (see text for details). The black lines show the relative WCB trajectory frequency at each profile height (contours at 1, 5, 10, 25, 50 and 100%), the red dashed line the temperature (contours at 0°, −20° and −40°C), and the brown line the 2-pvu contour, all three fields interpolated from ERA5.



**Figure 5.** Composite over all WCBs of the (a, b) median and (c-f) mean vertical profiles of ERA5 fields, separately for different height bins of the matching WCB air parcels. The shading shows (a) the sum of IWC and SWC (mg m$^{-3}$), (b) the sum of LWC and RWC (mg m$^{-3}$), (c) relative humidity with respect to ice (%), (d) $\theta_e$ (K), (e) moist vertical stability $d\theta_e/dz$ (K km$^{-1}$), and (f) PV (pvu). The black and brown contours are as in Fig. 4.

**Figure 6.** ERA5-based total (solid), large-scale (long-dashed) and convective (short-dashed) surface precipitation accumulated during the previous hour, averaged over all (green) and strong WCBs (blue), separately for different height bins of the matching WCB air parcels.



**Figure 7.** As Fig. 5, but for the category of strong WCBs.