# Peer review of "Vertical cloud structure of warm conveyor belts – a comparison and evaluation of ERA5 reanalyses, CloudSat and CALIPSO data"

_Weather and Climate Dynamics, 2020_

## Referee Comment (RC1) · Josué Gehring (Referee) · 31 Jul 2020

31/07/20

**Review of "Vertical cloud structure of warm conveyor belts – a comparison and evaluation of ERA5 reanalyses, CloudSat and CALIPSO data" by Hanin Binder, Maxi Boettcher, Hanna Joos, Michael Sprenger, and Heini Wernli**

**General comments**

The authors investigate the cloud and precipitation associated with WCBs during nine Northern Hemisphere winters with ERA5 reanalyses, CloudSat and CALIPSO data. They use ERA5 to not only depict the meteorological conditions associated with these WCB-produced cloud systems, but also to compare its performance with the measurements. They provide novel findings on the climatology of clouds and precipitation associated with WCBs and their corresponding thermodynamical and dynamical fields. In particular, the small- and mesoscale structures depicted with the satellite measurements are unprecedented. I strongly support the conclusions of this study and its publication in Weather and Climate Dynamics, subject to minor revisions.

**Specific comments**

L.29. "[...], which can intensify the associated cyclone (Binder et al., 2016)"
Maybe you could add Davis and Emanuel, 1991; Rossa et al., 2000.

L. 51, 54: Blanchard et al. 2020 could be added either on L. 51 or 54 along with Oertel et al. 2019 or 2020 respectively, since it is also an example of embedded convection in a WCB during NAWDEX, respectively an example of mesoscale PV dipoles.

L.106-107: The sensitivity of the CPR ranges from −30 to 50 dBZ.
This is quiet a large interval, could you give an indication of the sensitivity as a function of range (e.g. -30 dBZ at X m a.s.l., 50 dBZ at Y m a.s.l.)? I guess the sensitivity of 50 dBZ is in the case of attenuation?

L. 108: [...], which can amount to more than 10 dBZ km$^{-1}$ […]
Could you give a value or just a qualitative statement of the attenuation if no liquid water is present? Even at 94 GHz, the attenuation by gases and ice is much smaller than the one of liquid water. This would help a reader not familiar to radar attenuation to understand that it is less significant above the melting layer (respectively above the highest supercooled liquid water layer).

L. 159-170: I like the modifications to identify the WCB with respect to Madonna et al. (2014) method, I think it makes totally sense. However a 45% total increase of the number of trajectories is quiet substantial. Could you explain a bit the motivation behind these modifications? Is it motivated specifically by the aim of this study or should it apply to all future WCB detections? Do you expect your results to be sensitive to this increase in the number of trajectories? For instance, it could modify the distribution of the type of WCB included (e.g. more rapidly ascending ones with respect to Madonna et al. (2014) method).

Section 2.4: While the matching between WCB trajectories and CloudSat – CALIPSO overpasses is very well explained, I think a schematic of the method could help to visualise it, if feasible. Otherwise, referring to Fig. 1b could already help.

L. 179: "the 56 preceding and succeeding profiles" seemed to come out of nowhere the first time I read it. I then understood that 113 profiles times the horizontal resolution of DARDAR would make 124 km, but at first it is hard to understand the rational behind these numbers. Maybe it would help the reader to explain where the 56 comes from. This is somehow related to the previous comment. If you include a schematic on the matching, this could be added in it.

L. 239-247: I really like how you summarise the end of a section and introduce a new one.

L.273-275: Do you have a possible explanation for that?

L. 363: "[…], and there is no clear indication for a melting layer in the radar signal."
Actually if you look at the shape of the reflectivity contours (Fig. 4c), even if it is mainly oriented along the diagonal, there is a change in the curvature which follows the 0°C isotherm, indicating a secondary maximum of reflectivity. I think this could be an indication of the melting layer in the radar signal. To fully appreciate the melting layer signature at W-band, one has to consider the following: (i) For a ground-based W-band radar there is no bright band as is the case at lower frequencies, instead "an abrupt increase in the radar reflectivity without a following decrease at the base of the melting layer" (Kollias and Albrecht, 2005). (ii) For a nadir-pointing W-band radar (e.g. CloudSat) the bright band occurs du to "an increase in radar reflectivity from the dielectric effect of water, followed by a rapid signal decline [...] caused by correspondingly strong signal attenuation." (Sassen et al. 2007). I think both cases (all WCBs and strong WCBs) nicely correspond to the description of Sassen et al. 2007: we have a maximum of reflectivity followed by a sharp decrease due to attenuation (especially in Fig. 4c, which is consistent with stronger rainfall). It seems that the maximum of reflectivity is not due to the higher dielectric constant of the liquid water coating the melting snowflakes, since it occurs above the 0°C isotherm, but rather to attenuation starting above the melting layer. Hence, it suggests that the attenuation effect is stronger than the dielectric effect. The attenuation starting above the melting layer could be partially explained by the presence of supercooled liquid water droplets (see Fig. 5b and 7b): they are so tiny, that even at W-band they will not significantly contribute to the reflectivity, despite there higher dielectric constant than the surrounding snowflakes. However, they contribute significantly to the attenuation.

L. 434: Even above $z_{WCB}$=9km I see no gap between the frozen and liquid hydrometeors. Do I interpret this correctly?

Figure 2/5/7: Maybe it could be recalled in the caption that $IWC_{DARDAR}$ includes SWC and hence $IWC_{DARDAR}$ should be compared to $IWC_{ERA5}$ + $SWC_{ERA5}$. For a reader comparing Fig. 5 and 4 without reading Sect. 4.1, this could be confusing.

Figure 4c: There is a nice secondary maximum of reflectivity for $z_{WCB} >$ 12 km and $z_{prof}$ between 3 and 7 km. This is probably associated with the subtropical convective system mentioned in Sect. 4.1. It is also well depicted in Fig. 7a,b and seem to correspond to negative moist static stability (Fig. 7e). Even if subtropical systems are not the focus of this study, it could be mentioned, since it is a rather interesting feature.

**Technical corrections**

L. 61: According to WCD guidelines, footnotes should be avoided. I personally find it OK, it is just for you to know in case.

L. 92: According to WCD guidelines, Section should be abbreviated Sect. in running text, this should be corrected throughout the manuscript.

L. 175: Suggestion: "A match between a WCB trajectory position and the satellites' track [..]" would be maybe more precise.

L. 214: Suggestion: "[…], the lower reflectivity and IWC values suggest ice clouds rather than

falling snow." would be more appropriate than "indicate".

L. 225: "The IWC values strongly increase [...]" would be more precise than "The values strongly increase [...]"

L. 226: "[…] (see grey contours)." You mean the thin black contours of SWC in Fig. 2d?

L. 193, 245, 372: I am not sure if the capitalisation of "Section" is correct if it does not refer to a specific section (i.e. followed by a number)? To be checked.

L.413: you probably mean Fig. 5b.

L. 445: "In the WCB outflow, the PV values are anomalously low  (< 0.2 pvu)." The second "PV" seems not necessary.

L. 569: Just in case, this article is now in final form in ACP: https://doi.org/10.5194/acp-20-7373-2020

Figure 6: Why not using the same colour for all WCBs and strong WCBs as in Fig. 3d or the opposite?

Josué Gehring

**References**

Blanchard, N., Pantillon, F., Chaboureau, J.-P., and Delanoë, J.: Organization of convective ascents in a warm conveyor belt, *Weather Clim. Dynam. Discuss.*, https://doi.org/10.5194/wcd-2020-25, in review, 2020.

Davis, C. and Emanuel, K.A. (1991) Potential vorticity diagnostics of cyclogenesis. *Monthly Weather Review*, 119, 1929–1953.

Kollias, Pavlos, and Bruce Albrecht. "Why the Melting Layer Radar Reflectivity Is Not Bright at 94 GHz." *Geophysical Research Letters* 32, no. 24 (2005). https://doi.org/10.1029/2005GL024074.

Madonna, E., Wernli, H., Joos, H., and Martius, O.: Warm conveyor belts in the ERA-Interim dataset (1979–2010). Part I: climatology and potential vorticity evolution, *J. Climate*, 27, 3–26, https://doi.org/10.1175/JCLI-D-12-00720.1, https://doi.org/10.1175/JCLI-D-12-00720.1, 2014.

Rossa, A., Wernli, H. and Davies, H.C. (2000) Growth and decay of an extratropical cyclone's PV-tower. *Meteorology and Atmospheric Physics*, 73, 139–156.

Sassen, Kenneth, Sergey Matrosov, and James Campbell. "CloudSat Spaceborne 94 GHz Radar Bright Bands in the Melting Layer: An Attenuation-Driven Upside-down Lidar Analog." *Geophysical Research Letters* 34, no. 16 (2007). https://doi.org/10.1029/2007GL030291.

---

## Referee Comment (RC2) · Derek J. Posselt (Referee) · 11 Aug 2020

Review of Vertical cloud structure of warm conveyor belts - a comparison and evaluation of ERA5 reanalyses, CloudSat, and CALIPSO data

by H. Binder, M. Boettcher, H. Joos, M. Sprenger, and H. Wernli

Summary: This paper uses retrievals of ice water content from CloudSat and CALIPSO to study the properties of warm conveyor belts, and to evaluate the clouds produced by the ERA5 reanalysis. The authors analyze a representative WCB case, then extend their analysis to multiple years of observations, producing composites of retrieved ice

water content for all WCBs and for the 5% corresponding to the largest radar reflectivity values. They find interesting signals in the observations that indicate a connection between WCB properties and the vertical distribution of ice cloud content. They further connect the ice cloud properties to dynamics and thermodynamic properties by analyzing the static stability and PV in the reanalysis data. The paper is well conceived and well written, and I have only minor comments and suggestions for the authors. I begin with a few general comments, then itemize a number of specific comments after.

General Comments:

1. One must be careful, when comparing model output to satellite retrievals, to note where there may be overlap between the sources of data. In the case of DARDAR ice water content retrievals, the ice cloud estimates depend on temperature profiles that are obtained from ECMWF. As such, the IWC information in the retrievals is not entirely independent of the model that is being evaluated. This does not mean that the comparison is not valid, but I would suggest that the authors note the fact that there is information from ECMWF in the DARDAR product and perhaps discuss how this might affect the conclusions drawn in the study.

2. There appears to be a convective (vs stratiform) signal present for strong WCBs with heights above 12 km (see my specific comments below for details). I wonder if it would be of interest to the authors to add a brief discussion of this?

3. One question that comes to mind in any observation-based study (and in an evaluation of model output in particular) is the degree to which observations are able to characterize processes. The authors have done a nice job inferring the connection between satellite retrievals and processes via the analysis of PV and the thermodynamic environment. However, the A-Train sees only snapshots of cloud fields and (as the authors point out) rarely revisits a given storm more than once in its life cycle. Looking toward the future, I wonder what is missing from the observations that would enable a more specific process-based analysis and model evaluation? What are the most

critical observational needs for improving the understanding of the time evolution of storms? Will the addition of vertical motion estimates (e.g., from the doppler radar on EarthCARE) be helpful, or are time resolved measurements of the interior of the WCB clouds necessary? The paper is complete with or without this discussion, but I wonder if the authors would like to comment (given their expertise in this area) on what would be most useful in their future analyses?

Specific Comments:

1. P6, line 173. Is it true that the satellite observations are available more often than hourly? Even over the poles, the data is available only approximately once every 100 minutes, with longer delays with decreasing latitude. I wonder if you meant that the satellite intersections of WCBs often happen between the 1-hour analysis intervals?

2. P8, lines 227-229. What is the uncertainty in the DARDAR IWC retrieval? Is there a known issue with retrieval of ice in mixed phase regions in DARDAR? I wonder whether the differences between ERA-5 and DARDAR are within the 1-sigma uncertainty of the observations? The same question would apply to the composite comparisons in section 4.

3. P8, lines 249-250. In Figs 3a and 3b, I noticed an apparent gap in the concentration of WCBs (in the full set and also the "strong" subset) around 170 W longitude and extending through all latitudes from SE to NW. It looks like there may be data missing along a satellite swath? I am curious as to whether this is a real (geophysical) feature or an artifact in the data?

4. P11, lines 354-355. It is interesting that there are large reflectivities (and relatively warm columns) where the "strong" WCB heights are greater than 12 km. As the authors point out, it is not possible from Figs. 4c and 4d to determine trajectories; however, I wonder if the fact that the mean profiles extend through the depth of the troposphere with high reflectivities and warm temperatures indicate convective (vs stratiform) profiles?

5. P12, lines 377-378. I have the same question here as I did in for the case study - how do the obs-model differences compare with the observational uncertainty? In this case, would one expect the observation uncertainty to be smaller than for the individual case (since the results consist of an average over a large number of profiles)?

6. P14, lines 425ff. As noted above in my comment on the deep protions of strong WCBs in Fig. 4c, the thermodynamic analysis appears to support the presence of convective clouds for WCB trajectory heights above 11 km. There is a strong uptick in precipitation rate (including convective) (Fig. 6) as well as high RH through the depth of the troposphere (Fig. 7c), high theta-e values (Fig 7d), and weak stratification (Fig. 7e).

---

## Author Response (AR1)

Paper wcd-2020-26

"Vertical cloud structure of warm conveyor belts – a comparison and evaluation of ERA5 reanalyses, CloudSat and CALIPSO data"

by Hanin Binder, Maxi Boettcher, Hanna Joos, Michael Sprenger and Heini Wernli

**Response to the Reviewer's comments:**

We thank both reviewers for their constructive reviews that helped to improve our manuscript. We did our best to follow the suggestions. Essentially, we have made the following important changes:

- To ease the understanding, we have included a schematic illustration of the method to attribute several satellite profiles to a matching ERA5-based WCB trajectory in the climatological analysis (comment of reviewer 1).
- We have included a brief discussion of the radar dim band at the melting layer, which is characteristic for millimetre-wavelength radars like CloudSat (comment of reviewer 1).
- Thanks to a comment of reviewer 2, we found a data gap around 170°W and incorporated the missing data in the climatological analysis. The updated figures are almost identical to those in the previous version, such that the inclusion of the missing data does not affect the findings of this study.
- We have included a discussion of the uncertainties in the satellite-retrieved IWC values (comment of reviewer 2).

Below are the detailed replies to the individual comments.

**Reviewer 1 (Josué Gehring)**

**General comments**

*The authors investigate the cloud and precipitation associated with WCBs during nine Northern Hemisphere winters with ERA5 reanalyses, CloudSat and CALIPSO data. They use ERA5 to not only depict the meteorological conditions associated with these WCB-produced cloud systems, but also to compare its performance with the measurements. They provide novel findings on the climatology of clouds and precipitation associated with WCBs and their corresponding thermodynamical and dynamical fields. In particular, the small- and mesoscale structures depicted with the satellite measurements are unprecedented. I strongly support the conclusions of this study and its publication in Weather and Climate Dynamics, subject to minor revisions.*

**Specific comments**

*L.29. "[...], which can intensify the associated cyclone (Binder et al., 2016)"*
*Maybe you could add Davis and Emanuel, 1991; Rossa et al., 2000.*

**Reply:** Thank you, we now mention these studies as well.

*L. 51, 54: Blanchard et al. 2020 could be added either on L. 51 or 54 along with Oertel et al. 2019 or 2020 respectively, since it is also an example of embedded convection in a WCB during NAWDEX, respectively an example of mesoscale PV dipoles.*

**Reply:** This is true, we now mention Blanchard et al. 2020 along with Oertel et al. 2019.

*L.106-107: The sensitivity of the CPR ranges from −30 to 50 dBZ. This is quite a large interval, could you give an indication of the sensitivity as a function of range (e.g. -30 dBZ at X m a.s.l., 50 dBZ at Y m a.s.l.)? I guess the sensitivity of 50 dBZ is in the case of attenuation?*

**Reply:** We apologize, this sentence was confusing, we wanted to say that the minimum detectable reflectivity is -30 dBZ and the maximum 50 dBZ. However, we will now just mention the minimum, as the maximum is not important for our study. We changed the sentence to: "The minimum detectable signal of the CPR is -30 dBZ."

*L. 108: [...], which can amount to more than 10 dBZ km-1 [...]*
*Could you give a value or just a qualitative statement of the attenuation if no liquid water is present? Even at 94 GHz, the attenuation by gases and ice is much smaller than the one of liquid water. This would help a reader not familiar to radar attenuation to understand that it is less significant above the melting layer (respectively above the highest supercooled liquid water layer).*

**Reply:** Unfortunately, we could not find more quantitative information about attenuation in situations without liquid water. In order to emphasize the point that attenuation by liquid water is more relevant, we changed the sentence to "Absorption mainly by liquid water results in a two-way attenuation of the radar signal, which can amount to more than 10 dBZ km$^{-1}$ below the melting layer and even lead to a full attenuation of the radar signal in strongly precipitating systems (Mace et al., 2007; Marchand et al., 2008)."

*L. 159-170: I like the modifications to identify the WCB with respect to Madonna et al. (2014) method, I think it makes totally sense. However a 45% total increase of the number of trajectories is quite substantial. Could you explain a bit the motivation behind these modifications? Is it motivated specifically by the aim of this study or should it apply to all future WCB detections? Do you expect your results to be sensitive to this increase in the number of trajectories? For instance, it could modify the distribution of the type of WCB included (e.g. more rapidly ascending ones with respect to Madonna et al. (2014) method).*

**Reply:** These modifications with respect to the original method are not specifically motivated by the aim of this study, but they will be used in all our future studies based on ERA5. When calculating the new WCB climatology in ERA5 we took the chance to incorporate a few modifications that we consider to be meaningful, based on our experience with ERA-Interim. It is true that these modifications lead to a substantial increase in the number of identified trajectories, and in particular there are more rapidly ascending ones compared to the original criteria. We tested the sensitivity of the results of this study to the first of the two modifications,

i.e., to also select very fast ascending trajectories that fulfil the 600-hPa ascent criterion in the first part of the 48-h period and thereafter descend again. Fig. R1 is equivalent to Fig. 5 in the paper, but only includes WCB trajectories according to the Madonna et al. ascent criterion of 600 hPa between times 0 and 48 h (but with the new clustering criterion). Both for all and for the strong WCBs the observed reflectivity and IWC patterns are very similar to those in Fig. 4 in the paper where we used the modified ascent criterion. This is also the case for all other figures shown in the paper. For the second modification no sensitivity study has been done, but also here we expect the sensitivity to be relatively small.

A very detailed comparison between the WCB climatology based on ERA-Interim and ERA5, respectively, is currently done in our group by Katharina Heitmann. She assesses in detail the impact of model differences and the modifications in the WCB identification on the frequency, geographical distribution and characteristics of the WCBs. First preliminary results show that while the modified WCB identification leads to an increase in the frequency of WCBs, the geographical distribution and main characteristics of WCBs are very similar to those documented in Madonna et al. (2014). Since this is work in progress and not the focus of the present study, we prefer not to go into much detail about these technical issues in the present study. Due to the small sensitivity of our results to the exact WCB identification criterion, we regard it as justified to use the slightly modified criteria without discussing the motivation in much detail.

On lines 173-175 we added the sentence "Despite the significant increase in the number of identified WCB trajectories with respect to the original WCB identification criterion, tests have shown that the modifications do not affect the findings of this study."

[Figure]

Fig. R1: Equivalent to Fig. 5 in the manuscript, but without considering very fast ascending WCB trajectories that fulfil the 600-hPa ascent criterion in the first part of the 48-h period and thereafter descend again, i.e., when using the original Madonna et al. (2014) selection criteria for WCB trajectories.

*Section 2.4: While the matching between WCB trajectories and CloudSat – CALIPSO overpasses is very well explained, I think a schematic of the method could help to visualise it, if feasible. Otherwise, referring to Fig. 1b could already help.*

**Reply:** We now refer to Fig. 1b to illustrate the matching and include a schematic to illustrate how several satellite profiles are attributed to a single matching WCB trajectory in the climatological analysis (see reply below).

*L. 179: "the 56 preceding and succeeding profiles" seemed to come out of nowhere the first time I read it. I then understood that 113 profiles times the horizontal resolution of DARDAR*

*would make 124 km, but at first it is hard to understand the rationale behind these numbers. Maybe it would help the reader to explain where the 56 comes from. This is somehow related to the previous comment. If you include a schematic on the matching, this could be added in it.*

**Reply:** You are right, we now include a schematic to illustrate the method (see Fig. R2 below and Fig. 1 in the revised manuscript) and tried to improve the explanation by slightly changing the sentences as follows:

"In the climatological study, several satellite profiles are attributed to each matching WCB trajectory. They are then averaged to increase the representativity of the observations (see schematic illustration in Fig. 1). In total, 113 satellite profiles are averaged per WCB-match - that is, the profile with the closest distance to the WCB air parcel, plus the 56 preceding and the 56 succeeding profiles. With a distance of 1.1 km between each satellite profile, the 113 assigned profiles correspond to a track segment length of about 124 km …"

[Figure]

Fig. R2: Schematic illustration of the method to attribute several satellite profiles to a matching ERA5-based WCB trajectory in the climatological analysis.

*L. 239-247: I really like how you summarise the end of a section and introduce a new one.*

**Reply:** Thank you!

*L.273-275: Do you have a possible explanation for that?*

**Reply:** We are not sure why WCBs with exceptionally strong radar reflectivities in their inflow occur further north than the mean over all matches. We have to leave it for further research to better understand this behaviour.
However, we would like to add one possible hint, which is based on an unpublished earlier analysis. For these older results, Fig. R3 shows horizontal composites of the upper-level PV structure for all and the strong WCBs in the inflow (i.e., for matches below 2.5 km height). The coordinates are defined relative to the position of the matching WCB trajectory, which is relocated to 0° longitude and latitude. Both all and the strong WCBs are located close to the extratropical upper-level waveguide between a trough and a ridge. However, the strong WCBs are located about 5° closer to the waveguide, right at the leading edge of the trough, and the PV gradient associated with the trough is stronger. This suggests that the further northward located inflow in the subcategory of strong WCBs experiences a stronger upper-level forcing for ascent, which could explain the particularly strong radar reflectivities associated with them.

The lower tropopause and accordingly the stronger upper-level forcing for ascent in the subcategory of strong WCBs are also evident in the vertical composites in the manuscript (compare Figs. 6f and 8f). However, this possible explanation is rather speculative, and we prefer not to include it in the paper.

[Figure]

Fig. R3: Composites of PV at 320 K (pvu; shading) and sea level pressure (black contours every 3 hPa) over (left) all WCBs and (right) strong WCBs that are located below 2.5 km height. The black cross at the centre marks the position of the matching WCB trajectories.

*L. 363: "[...], and there is no clear indication for a melting layer in the radar signal."*
*Actually if you look at the shape of the reflectivity contours (Fig. 4c), even if it is mainly oriented along the diagonal, there is a change in the curvature which follows the 0°C isotherm, indicating a secondary maximum of reflectivity. I think this could be an indication of the melting layer in the radar signal. To fully appreciate the melting layer signature at W-band, one has to consider the following: (i) For a ground-based W-band radar there is no bright band as is the case at lower frequencies, instead "an abrupt increase in the radar reflectivity without a following decrease at the base of the melting layer" (Kollias and Albrecht, 2005). (ii) For a nadir-pointing W-band radar (e.g. CloudSat) the bright band occurs du to "an increase in radar reflectivity from the dielectric effect of water, followed by a rapid signal decline [...] caused by correspondingly strong signal attenuation." (Sassen et al. 2007). I think both cases (all WCBs and strong WCBs) nicely correspond to the description of Sassen et al. 2007: we have a maximum of reflectivity followed by a sharp decrease due to attenuation (especially in Fig. 4c, which is consistent with stronger rainfall). It seems that the maximum of reflectivity is not due to the higher dielectric constant of the liquid water coating the melting snowflakes, since it occurs above the 0°C isotherm, but rather to attenuation starting above the melting layer. Hence, it suggests that the attenuation effect is stronger than the dielectric effect. The attenuation starting above the melting layer could be partially explained by the presence of supercooled liquid water droplets (see Fig. 5b and 7b): they are so tiny, that even at W-band they will not significantly contribute to the reflectivity, despite their higher dielectric constant than the surrounding snowflakes. However, they contribute significantly to the attenuation.*

**Reply:** Many thanks for these useful insights, they have greatly helped us to better understand the signal at the melting layer. You are right that the description of Sassen et al. (2007) fits well with the reflectivity signal in the case study (Fig. 3a) and in the composite of strong WCBs (Fig. 5c): There is a first reflectivity maximum above the melting layer in the mixed-phase clouds, followed by a dim band at the top of the melting layer that is most likely due to snowfall attenuation. Below the melting level, a second increase in the signal is present that is probably caused by the dielectric effect of water, followed by a rapid signal decline due to rainfall attenuation.

In the revised manuscript, we briefly discuss the dim band in the case study on lines 223-227, and in the climatological part we deleted the words "and there is no clear indication for a melting layer in the radar signal" and now write instead that "the signal decrease below the WCB indicates strong snow and rain attenuation" (line 388).

*L. 434: Even above zWCB=9km I see no gap between the frozen and liquid hydrometeors. Do I interpret this correctly?*

**Reply:** You are right, thank you for pointing this out. We changed the sentence to "In contrast to all WCB matches, there is no gap between the frozen and liquid hydrometeors throughout the inflow, ascent and outflow (Fig. 7a,b), consistent with the higher $RH_{ice}$ (Fig. 7c)."

*Figure 2/5/7: Maybe it could be recalled in the caption that IWCDARDAR includes SWC and hence IWCDARDAR should be compared to IWCERA5 + SWCERA5. For a reader comparing Fig. 5 and 4 without reading Sect. 4.1, this could be confusing.*

**Reply:** Thank you for pointing this out, we now mention it explicitly in the captions of Fig. 3, 5 and 6. Furthermore, in each figure we changed the title of the label bar from "IWC" or "IWC + SWC" to "Frozen hydrometeor content", to make it clearer that IWC + SWC from ERA5 should be compared to IWCDARDAR.

*Figure 4c: There is a nice secondary maximum of reflectivity for zWCB> 12 km and zprof between 3 and 7 km. This is probably associated with the subtropical convective system mentioned in Sect. 4.1. It is also well depicted in Fig. 7a,b and seem to correspond to negative moist static stability (Fig. 7e). Even if subtropical systems are not the focus of this study, it could be mentioned, since it*
*is a rather interesting feature.*

**Reply:** It is true, this is an interesting feature as well, which has also been mentioned by reviewer 2. We now mention it on lines 391-393 and 474-478.

**Technical corrections**

*L. 61: According to WCD guidelines, footnotes should be avoided. I personally find it OK, it is just for you to know in case.*

**Reply:** Thanks, we now incorporate the footnotes in the main text.

*L. 92: According to WCD guidelines, Section should be abbreviated Sect. in running text, this should be corrected throughout the manuscript.*

**Reply:** Thanks, we changed it everywhere.

*L. 175: Suggestion: "A match between a WCB trajectory position and the satellites' track [..]" would be maybe more precise.*

**Reply:** You are right, thank you, we changed it as you suggested.

*L. 214: Suggestion: "[…], the lower reflectivity and IWC values suggest ice clouds rather than falling snow." would be more appropriate than "indicate".*

**Reply:** Thank you, we replaced "indicate" by "suggest".

*L. 225: "The IWC values strongly increase [...]" would be more precise than "The values strongly increase [...]"*

**Reply:** Thank you for pointing this out, it is true that we should be more precise here. It is the sum over the IWC and SWC values that strongly increases toward the melting layer, so we changed the sentence as follows: "The sum over the ice and snow values strongly increases toward the melting layer and peaks right above the melting layer in the deep frontal clouds, where most of it is present as falling snow (see thin black contours in Fig. 3d)."

*L. 226: "[…] (see grey contours)." You mean the thin black contours of SWC in Fig. 2d?*

**Reply:** You are right, we meant the thin black contours, we changed it accordingly (see comment above).

*L. 193, 245, 372: I am not sure if the capitalisation of "Section" is correct if it does not refer to a specific section (i.e. followed by a number)? To be checked.*

**Reply:** Thank you, we leave this to the copy editor.

*L.413: you probably mean Fig. 5b.*

**Reply:** Yes, thank you for spotting this typo.

*L. 445: "In the WCB outflow, the PV values are anomalously low PV (< 0.2 pvu)." The second "PV" seems not necessary.*

**Reply:** You are right, thank you.

*L. 569: Just in case, this article is now in final form in ACP: https://doi.org/10.5194/acp-20-7373-2020*

**Reply:** Thank you, we changed it accordingly.

*Figure 6: Why not using the same colour for all WCBs and strong WCBs as in Fig. 3d or the opposite?*

**Reply:** Good point, we changed the colours in Fig. 3d, they are now as in Fig. 6.

**Reviewer 2 (Derek Posselt)**

**Summary:** *This paper uses retrievals of ice water content from CloudSat and CALIPSO to study the properties of warm conveyor belts, and to evaluate the clouds produced by the ERA5 reanalysis. The authors analyze a representative WCB case, then extend their analysis to multiple years of observations, producing composites of retrieved ice water content for all WCBs and for the 5% corresponding to the largest radar reflectivity values. They find interesting signals in the observations that indicate a connection between WCB properties and the vertical distribution of ice cloud content. They further connect the ice cloud properties to dynamics and thermodynamic properties by analyzing the static stability and PV in the reanalysis data. The paper is well conceived and well written, and I have only minor comments and suggestions for the authors. I begin with a few general comments, then itemize a number of specific comments after.*

**General Comments:**

*1. One must be careful, when comparing model output to satellite retrievals, to note where there may be overlap between the sources of data. In the case of DARDAR ice water content retrievals, the ice cloud estimates depend on temperature profiles that are obtained from ECMWF. As such, the IWC information in the retrievals is not entirely independent of the model that is being evaluated. This does not mean that the comparison is not valid, but I would suggest that the authors note the fact that there is information from ECMWF in the DARDAR product and perhaps discuss how this might affect the conclusions drawn in the study.*

**Reply:** You are right, it is important to point out that the two datasets are not entirely independent, thank you for this comment. In particular, the use of temperature profiles from ECMWF analyses most likely explains the good agreement of the melting layer height in the satellite retrievals and ERA5, i.e., a good anchoring at the melting layer (Fig. 3c, 5b,d).
We now mention and discuss this on lines 121-123, 238-243 and 533-537.

*2. There appears to be a convective (vs stratiform) signal present for strong WCBs with heights above 12 km (see my specific comments below for details). I wonder if it would be of interest to the authors to add a brief discussion of this?*

**Reply:** We agree that there appears to be a convective signal, and it is most likely related to subtropical cyclones, because matches with strong WCBs at $z_{WCB} > 12$ km occur in the subtropics, at about 15-30°N (see blue line in Fig. 4d). We didn't include a discussion of it in the previous draft, as subtropical convective systems are not the focus of our study. But it is true that it is nevertheless an interesting feature that has also been noted by the other reviewer. We now include a brief discussion of it on lines 391-393 and 474-478.

*3. One question that comes to mind in any observation-based study (and in an evaluation of model output in particular) is the degree to which observations are able to characterize processes. The authors have done a nice job inferring the connection between satellite retrievals and processes via the analysis of PV and the thermodynamic environment. However, the A-Train sees only snapshots of cloud fields and (as the authors point out) rarely revisits a given storm more than once in its life cycle. Looking toward the future, I wonder what is missing from the observations that would enable a more specific process-based analysis and model evaluation? What are the most critical observational needs for improving the understanding of the time evolution of storms? Will the addition of vertical motion estimates (e.g., from the doppler radar on EarthCARE) be helpful, or are time resolved measurements of the interior of the WCB clouds necessary? The paper is complete with or without this discussion, but I wonder if the authors would like to comment (given their expertise in this area) on what would be most useful in their future analyses?*

**Reply:** This is a very interesting question. As you point out, the vertical motion estimates from the Doppler radar on the EarthCare satellite will certainly be very valuable to characterize the dynamical processes associated with WCBs in further detail, and to evaluate them in model data. Also, a higher spatial coverage would be useful, such that the temporal evolution of the satellite measurements along individual WCB trajectories could be investigated. While a full coverage will certainly not be feasible in the near future, the increase in the spatial coverage of radar and lidar data with the launch of the EarthCare satellite will already enable a more in-depth study of the time evolution of the WCBs and the associated cyclones.
As discussed in the ECMWF newsletter article of Rodwell et al. (2018), it is also important to accurately initialise humidity in models to improve the representation of WCBs, but it is challenging to do so. For instance, in cloudy regions such as WCBs, there is a lack of humidity measurements, as radiosondes are scarce, and water vapour lidars are strongly attenuated in clouds. Observational data of water vapor in cloudy regions would be very valuable for a better understanding of the cloud processes in WCBs.

**Specific Comments:**

*1. P6, line 173. Is it true that the satellite observations are available more often than hourly? Even over the poles, the data is available only approximately once every 100 minutes, with longer delays with decreasing latitude. I wonder if you meant that the satellite intersections of WCBs often happen between the 1-hour analysis intervals?*

**Reply:** We apologize, this was not well explained in the manuscript. It is true that the temporal resolution of the satellite data at one location is very low, with a repeat cycle of 16 days. However, we would like to say that the individual satellite profiles are available every about

0.16 seconds (without referring to a specific location) and therefore much more often than the hourly ERA5 data. We reformulated this on lines 181-182.

*2. P8, lines 227-229. What is the uncertainty in the DARDAR IWC retrieval? Is there a known issue with retrieval of ice in mixed phase regions in DARDAR? I wonder whether the differences between ERA-5 and DARDAR are within the 1-sigma uncertainty of the observations? The same question would apply to the composite comparisons in section 4.*

**Reply**: Figure R4 shows for the case study the DARDAR-retrieved IWC (same as Fig. 3e in the paper) and the associated uncertainty in a running mean over 113 satellite profiles. The uncertainties are expressed as one standard deviation percentage errors in the natural logarithm of the IWC. For instance, an uncertainty of 50% for an IWC value of 2 mg m$^{-3}$ implies that the value ranges between 2 mg m$^{-3}$ / 1.5 and 2 mg m$^{-3}$ x 1.5 (for details see Eliasson et al. 2013: Systematic and random errors between collocated satellite ice water path observations. J. Geophys. Res.)

The uncertainties are about 20-30% in the pure ice clouds at upper levels, where the IWC values are low. Between the -20°C isotherm and the melting layer, where the IWC values are high and presumably associated with mixed-phase clouds, the uncertainties range from 5-20% near -20°C and 10-40% near the melting layer. Even with uncertainties of 40%, the difference between the maximum IWC values in the observations (1640 mg m$^{-3}$) and ERA5 (1050 mg m$^{-3}$) are larger than the one-sigma uncertainty range of the observations. This is also the case for the composite comparison, where the uncertainties are significantly lower than in the case study because of the averaging over a large number of satellite profiles (see reply to question 5).

We do not know of an issue with the retrieval of ice in the mixed-phase region in DARDAR. However, the patterns in Figs. R4b and R6b,d (see reply to question 5) indicate that the uncertainties are indeed highest in the lower part of the mixed-phase clouds near the melting layer.

We now include a discussion of the uncertainties in the DARDAR-retrieved IWC on lines 130-133, 249-252, 408-413 and 461-464.

[Figure]

Fig. R4: Vertical cross section along a North Pacific WCB at 00 UTC 3 January 2014 of (a) the DARDAR-retrieved IWC (mg m$^{-3}$; shading) and (b) the associated percentage uncertainty (%; shading) in a running mean along 113 satellite profiles, together with the ERA5-based temperature (red dashed contours at 0°, -20° and -40°C) and the 2-pvu contour (thick black line). (a) is equivalent to Fig. 3e in the paper.

*3. P8, lines 249-250. In Figs 3a and 3b, I noticed an apparent gap in the concentration of WCBs (in the full set and also the "strong" subset) around 170 W longitude and extending through all latitudes from SE to NW. It looks like there may be data missing along a satellite swath? I am curious as to whether this is a real (geophysical) feature or an artifact in the data?*

**Reply:** Thank you very much for noting this gap, there was indeed data missing along the satellite swath. We found the source of the problem and incorporated the missing data in the analysis. We updated all climatological figures, redid the analysis for the category of strong WCBs and changed the text accordingly. Compared to the previous version, the number of matches has increased from 502'977 to 509'042. This relatively small increase has almost no impact on the results and the updated figures are very similar to those in the previous version, such that incorporation of the previously missing data does not affect the findings of this study.

Note that in the updated Fig. 4a there is still an apparent gap around 165°W along a narrow region. Figure R5 shows the number of satellite profiles at each grid point that have been taken into account in the present study, together with the climatological WCB trajectory frequency for winter 2006 to 2016, i.e., for the considered years (the WCB trajectory frequency slightly differs from the one shown in Fig. 4a in the paper, where all years between 1980 and 2018 have been taken into account). The satellite data does not show any gaps, and also the WCB trajectory frequency is high around 165°W. This suggests that the apparent gap around 165°W in Fig. 4a is not an artefact of the data, but that by chance the number of WCB trajectories present in this region was slightly reduced during the satellite overpasses.

[Figure]

Fig. R5: Spatial distribution of the satellite profiles taken into account in the present study (shading). The colours show the number of profiles in each 0.5° bin. Overlaid is the ERA5-based climatological frequency of WCB trajectories for December-February 2006-2016, i.e., for the years considered in this study (blue contours every 10%). Note that the WCB trajectory frequency slightly differs from the one in Fig. 4a,b, which is shown for December-February 1980-2018.

*4. P11, lines 354-355. It is interesting that there are large reflectivities (and relatively warm columns) where the "strong" WCB heights are greater than 12 km. As the authors point out, it is not possible from Figs. 4c and 4d to determine trajectories; however, I*

*wonder if the fact that the mean profiles extend through the depth of the troposphere
with high reflectivities and warm temperatures indicate convective (vs stratiform) profiles?*

**Reply:** We agree that this is most likely a convective signal, and we now discuss it briefly on lines 391-393 and 474-478 (see also reply to the general comment 2).

*5. P12, lines 377-378. I have the same question here as I did in for the case study - how do the obs-model differences compare with the observational uncertainty? In this case, would one expect the observation uncertainty to be smaller than for the individual case (since the results consist of an average over a large number of profiles)?*

**Reply:** Figure R6 shows for all and the strong WCBs vertical composites of the percentage error of the IWC retrievals, separately for different WCB heights. Hereby, the mean error at each profile position has been divided by $\sqrt{N}$, i.e., the square root of the number of WCB matches in the respective WCB height bin (see histogram in Fig. 4c in the manuscript for the number of matches $N$ per WCB height). The observational error is therefore indeed much smaller than in the case study (cf. Fig. R4b).
For the entire climatology, the percentage error is particularly low, with values between 0.025% and 0.17% up to about $z_{WCB} \sim 12$ km, and slightly larger values of up to 1.7% for $z_{WCB}$ > 12 km (Fig. R6b). The larger errors at $z_{WCB}$ > 12 km are due to the much lower number of matches at this height (see histogram in Fig. 4c in the manuscript). At lower $z_{WCB}$, the error pattern strongly resembles the IWC pattern in Fig. R6a, with maximum errors of 0.17% at $z_{WCB}$ = 3.5-4 km and $z_{prof}$ = 3.5-4 km coinciding with the satellite-retrieved IWC maximum of 260 mg m$^{-3}$ (Fig. R6a). Thus, the error range is 260 mg m$^{-3}$ +/- 0.17%, i.e., the one-sigma uncertainty lies between 260 / 1.0017 = 259.6 mg m$^{-3}$ and 260 x 1.0017 = 260.4 mg m$^{-3}$. The ERA5-based maximum IWC values of 160 mg m$^{-3}$ are therefore significantly lower than the one-sigma percentage uncertainty of the observations, i.e., the conclusions in this study are valid.
In the category of strong WCBs, the errors are larger than for the entire climatology because of the smaller number of matches, but they are still significantly smaller than in the case study (Fig. R6d). For $z_{WCB}$ between ~0.5 and 12 km height, the error pattern again coincides approximately with the IWC pattern (Fig. R6c), with maximum values of 1-1.25% around the IWC maximum between the melting layer and the -20°C isotherm, and slightly lower values at higher altitudes. Hence, also for the category of strong WCBs the differences between the peak DARDAR-retrieved IWC values (1180 mg m$^{-3}$) and ERA5 (540 mg m$^{-3}$) are significantly larger than the one-sigma uncertainty range of the observations.

We now include a discussion of the uncertainties in the DARDAR-retrieved IWC on lines 130-133, 249-252, 408-413 and 461-464.

[Figure]

Fig. R6: Composites of (a, c) the DARDAR-retrieved IWC (mg m$^{-3}$; shading) and (b, d) the associated percentage uncertainty (%; shading), separately for different height bins of the matching WCB air parcels. The composites are shown for all WCBs in the top panels and for the strong WCBs in the bottom panels. The black lines show the relative WCB trajectory frequency at each profile height (contours at 1, 5, 10, 25, 50 and 100%), the red dashed line the temperature (contours at 0°, -20° and -40°C), and the brown line the 2-pvu contour, all three fields interpolated from ERA5. (a) and (c) are equivalent to Figs. 5b and 5d in the paper.

*6. P14, lines 425ff. As noted above in my comment on the deep portions of strong WCBs in Fig. 4c, the thermodynamic analysis appears to support the presence of convective clouds for WCB trajectory heights above 11 km. There is a strong uptick in precipitation rate (including convective) (Fig. 6) as well as high RH through the depth of the troposphere (Fig. 7c), high theta-e values (Fig 7d), and weak stratification (Fig 7e).*

**Reply:** It is true that the thermodynamic analysis supports the presence of convection below the WCB outflow at $z_{WCB}$> ~12 km, and we now discuss it briefly on lines 391-393 and 474-478 (see also reply to the general comment 2).

[revised manuscript text omitted]